# IPMix: Label-Preserving Data Augmentation Method for Training Robust Classifiers

**Zhenglin Huang**[1], **Xianan Bao**[1], **Na Zhang**[1]*, **Qingqi Zhang**[2],
**Xiaomei Tu**[3], **Biao Wu**[1], and **Xi Yang**[4]

[1]School of Artificial Intellienge, Zhejiang Sci-Tech University
[2]Yamaguchi University  [3]ZGUC
[4]University of Science and Technology of China, Hefei, China

## Abstract

Data augmentation has been proven effective for training high-accuracy convolutional neural network classifiers by preventing overfitting. However, building deep neural networks in real-world scenarios requires not only high accuracy on clean data but also robustness when data distributions shift. While prior methods have proposed that there is a trade-off between accuracy and robustness, we propose IPMix, a simple data augmentation approach to improve robustness without hurting clean accuracy. IPMix integrates three levels of data augmentation (image-level, patch-level, and pixel-level) into a coherent and label-preserving technique to increase the diversity of training data with limited computational overhead. To further improve the robustness, IPMix introduces structural complexity at different levels to generate more diverse images and adopts the random mixing method for multi-scale information fusion. Experiments demonstrate that IPMix outperforms state-of-the-art corruption robustness on CIFAR-C and ImageNet-C. In addition, we show that IPMix also significantly improves the other safety measures, including robustness to adversarial perturbations, calibration, prediction consistency, and anomaly detection, achieving state-of-the-art or comparable results on several benchmarks. Code is available at `https://github.com/hzlsaber/IPMix`.

## 1 Introduction

Deep neural network models have recently achieved remarkable performance on various computer vision tasks, such as zero-shot image classification [1–3], 3D object detection [4–6], and face recognition [7, 8]. In real-world scenarios, models can achieve impressive accuracy when training and test distributions are identical, but challenges appear when confronted with out-of-distribution examples [9–11], such as natural corruptions [12], adversarial perturbations [13], and anomaly patterns [14], necessitating robustness across distribution shifts. Data augmentation has been proposed to partially alleviate this issue, which applies diverse transformations on clean images to generate new training examples [15, 16]. Furthermore, a high diversity of augmented images enables neural networks to resist data distribution shifts and improve robustness [17]. Data augmentation approaches generally fall into three subgroups: image-level, patch-level, and pixel-level augmentations.

Image-level augmentation techniques [18–20] apply transformations on the whole image, such as brightness, sharpness, and solarization, to increase the total amount of training data. Patch-level augmentation techniques [21, 22] typically mask or replace a region of an image, compelling classifiers to focus on less discriminative portions. Meanwhile, pixel-level augmentation techniques [23, 24] mix images using pixel-wise weighted averages to increase diversity within the training dataset.

---

*Corresponding author

37th Conference on Neural Information Processing Systems (NeurIPS 2023).

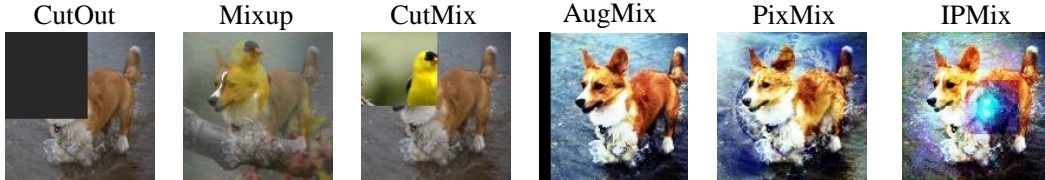

| CutOut | Mixup | CutMix | AugMix | PixMix | IPMix |

Figure 1: Visual comparison of various data augmentation methods. IPMix utilizes the structural complexity of fractals and multi-scale information to generate more diverse examples.

Previous studies have focused on either pixel-level or patch-level information to improve model performance. However, most of these techniques are label-variant, which may lead to manifold intrusion [25, 26] and decrease performance on unseen data. Simultaneously, a limitation of image-level data augmentation techniques is the computationally expensive search for an optimal augmentation policy, often exceeding the training process's complexity [18, 19]. Given these considerations and the potential for enhancing data augmentation strategies, we mainly discuss one question in this paper: *How to take advantage of the strengths of the three methods while avoiding their drawbacks?*

**Our contributions are as follows:**

- We propose **IPMix**, a label-preserving data augmentation approach, which integrates three levels of data augmentation into a single framework with limited computational overhead, demonstrating that these approaches are complementary and that a unification among them is necessary to achieve robustness.

- To further enhance model performance, IPMix incorporates structural complexity from synthetic data at various levels to produce more diverse images. Additionally, we employ random mixing methods and scar-like image patches for multi-scale information fusion.

- Extensive experiments demonstrate that IPMix achieves state-of-the-art corruption robustness and improves numerous safety metrics compared with other data augmentation approaches.

IPMix integrates the three data augmentation techniques in a label-preserving fashion, effectively circumventing potential manifold intrusion and maintaining label consistency[27]. Furthermore, inspired by prior work, IPMix eliminates the need to search for an optimal data augmentation policy, thus reducing computational costs. By addressing these challenges, IPMix has achieved significant improvements, as depicted in Figure 2. In comparison to other methods that focus on leveraging one of these categories for enhancement, IPMix achieves state-of-the-art results in accuracy and robustness.

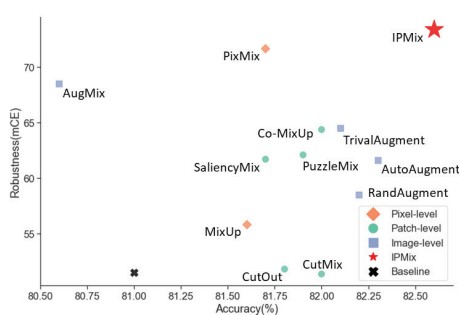

Figure 2: The performance of different levels of data augmentation methods on CIFAR-100. Compared to other approaches which focus on utilizing only one category, IPMix achieves state-of-the-art accuracy and robustness.

Since IPMix involves different levels of data augmentation techniques, it naturally motivates us to design a novel mixing method for better information fusion. Previous research has demonstrated that enhancing training data diversity [23, 28, 29] and image structural complexity [30, 31] is crucial for improving model robustness. The structural complexity of synthetic data, such as fractals and statistical information, can bolster model performance through pre-training [32] or blending with clean images [24]. For better data integration, IPMix mixes clean images with synthetic pictures at different scales by random mixing to improve structural complexity, which can generate more diverse images to improve robustness.

Building on the enhancement of corruption robustness, we further extend IPMix's capabilities to enhance various safety metrics to fulfill the demands of constructing secure and reliable systems in real-world situations [11]. We demonstrate that IPMix improves numerous safety metrics, including corruption robustness, calibrated uncertainty estimates, adversarial robustness, anomaly detection, and prediction consistency. On CIFAR-10-C and CIFAR-100-C, IPMix achieves the best results across different architectures. On ImageNet, IPMix outperforms previous methods and gains a substantial improvement on various safety measure benchmarks, achieving state-of-the-art or comparable results on ImageNet-R, ImageNet-A, and ImageNet-O [33, 34].

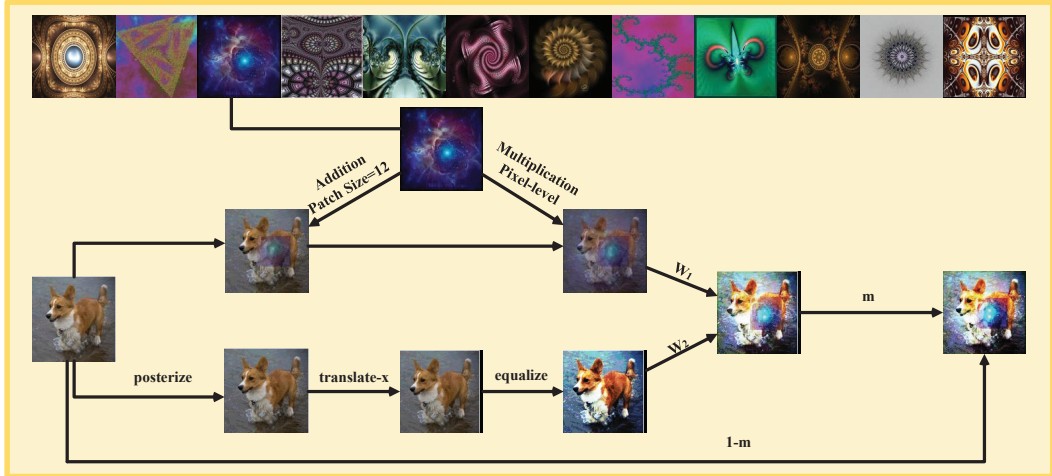

Figure 3: Top: Sample fractals from IPMix set. Bottom: An example of IPMix applied on a dog image, $k = 2$, $t = 3$. We randomly select P (pixel and patch) data augmentation methods and image-level data augmentation methods to generate a highly diverse set of augmented images. We sample $w_k$ ($k = 2$, in this case) from Dirichlet distribution and use skip connection ($m$ sample from a Beta distribution) to maintain semantic consistency.

## 2 Related Works

### 2.1 Data Augmentation

Data augmentation is crucial to the success of modern neural networks, contributing significantly to the improvement of model generalization performance. The presented data augmentation approaches can be classified into three high-level categories: image-level, pixel-level, and patch-level augmentations.

**Image-level data augmentation.** Image-level data augmentation methods are commonly label-preserving, applying transformations on the whole image to improve data diversity. AutoAugment [19] utilizes reinforcement learning to automatically search optimal compositions of transformations. Adversarial AutoAugment [35] generates adversarial images to extend data and produces a dynamic policy during training. TrivialAugment [36] randomly selects an operation and the magnitude to reduce search space and improve performance. AugMix [37] uses multiple transformations to create high diversity of augmented images, achieving state-of-the-art results on corruption robustness and calibration. AugMax [29] unifies diversity and hardness to search for the worst-case mixing strategy. PRIME [38] uses max-entropy image transformations to boost model corruption robustness.

**Pixel-level data augmentation.** Pixel-level data augmentation methods mix images using pixel-wise weighted averages. MixUp [23] generates augmented images by linearly interpolating between two randomly selected images and their corresponding labels. Manifold MixUp [39] encourages neural networks to learn smooth interpolations between data points in the hidden layers, improving accuracy by comparison with MixUp. PixMix [24] utilizes structural complexity synthetic pictures, such as fractals and feature visualizations, to improve model performance. Our work shared similarities with PixMix, but we use multi-scale information and better information fusion methods to train robust models by leveraging more diverse examples.

**Patch-level data augmentation.** Patch-level data augmentation methods mask or replace parts of the original image with different information. CutOut [28] randomly masks out regions of a clean image to learn less discriminative portions, thereby improving accuracy. CutMix [40] replaces a patch of an original image with another randomly picked image to improve performance. Patch Gaussian [41], which inputs a patch of Gaussian noise into the clean picture, combines the improved accuracy of CutOut with the noise robustness of Gaussian. SaliencyMix [42], based on the maximum intensity pixel local in the saliency map, replaces a square patch of the original image with salient information from another image. TokenMix [43] improves the performance of vision transformers by partitioning the mixing region into multiple separated parts and mixing two images at the token level. AutoMix [44] optimizes both the mixed sample generation task and the mixup classification task in a momentum training pipeline with corresponding sub-networks in a bi-level optimization framework.

## 2.2 Safety Measures

When deploying network models in real-world scenarios, it is crucial to consider comprehensive security measures beyond standard accuracy. Implementing unsafe machine learning systems in high-stakes environments [45–47] can lead to incalculable losses. With the rise of multimodal large language models (MLLMs) [48–50], safety issues are receiving increasing attention because their superior performance still makes mistakes. For example, GPT-4 [49] may be confidently wrong in its predictions and disturbed by adversarial questions. Previous research has proposed various safety measures, including but not limited to robustness and calibration.

**Robustness.** Corruption robustness considers how to improve the model resistance to unseen natural perturbations under data distribution shifts. As a variant of the original ImageNet, ImageNet-C [51] consists of 15 diverse commonplace corruptions belonging to different categories with five levels of severity, regarded as a general benchmark for corruption robustness. In addition to natural corruption, Hendrycks et al. [33] demonstrate that models should measure generalization to various abstract visual renditions. The robustness of adversarial attacks focuses on defending against imperceptible perturbations to images [52]. Prior works have proposed that there is a trade-off between the robustness of adversarial perturbations and clean image accuracy [53, 54]. ImageNet-O and ImageNet-A [34], widely regarded as benchmarks for evaluating image classifier performance under shifts in both input data and label distributions, are utilized for anomaly detection.

**Calibration.** Calibrated prediction confidences, which indicate whether a model's output should be trusted, are valuable for classification models in real-world settings. Bayesian approaches [55] are widely used to deal with uncertainty estimation. Kuleshov et al. [56] utilize recalibration methods to solve the miscalibration of credible intervals. Ovadia et al. [57] provide a benchmark for evaluating the accuracy and uncertainty of models under data distributional shifts.

## 2.3 Training with Synthetic Data

Previous works have proved that training with synthetic data can improve performance on real datasets. Debidatta et al. [58] discover that combining synthetic annotated datasets with real data can significantly improve the performance of instance detection. Baradad et al. [31] generate synthetic data by utilizing various procedural noise models. In addition, they find that naturalism and diversity are two important properties for synthetic data to achieve comparable results with real datasets. Kataoka et al. [59, 60] propose a suite of datasets generated by formula-driven supervised learning.

## 3 An Attempt to Integrate Existing Approaches

Some prior studies [24, 40] have suggested that combining different data augmentation techniques with existing methods can improve accuracy on standard datasets. However, these works merely employed simple combinations without considering the compatibility between methods at different levels. Simultaneously, these studies chose the clean accuracy as the sole evaluation metric and have not taken the model's

Table 1: The combination of different levels of data augmentation. M, C and A are abbreviations for MixUp, CutMix, and AugMix, respectively.

|  | Classification Error($\downarrow$) | Robustness mCE($\downarrow$) | Calibration RMS($\downarrow$) |
|---|---|---|---|
| Vanilla | 21.3 | 50 | 14.6 |
| +M | 20.5 (-0.8) | 45.9(-4.1) | 10.5(-4.1) |
| +M+C | 20.2 (-1.1) | 46.1(-3.9) | 22.7(+8.1) |
| +M+C+A | 23.4 (+2.1) | 50.1(+0.1) | 25.6(+11) |

safety performance into account. In this section, we select MixUp [23], CutMix [40], and AugMix [37] as representative data augmentation approaches for pixel-level, patch-level, and image-level, respectively, to conduct combination experiments of these approaches on CIFAR-100. Please refer to Appendix F for more details about the combination experiments.

Results on Table 1 demonstrate that simply combining different data augmentation methods may significantly impair model performance. This could be attributed to the excessive perturbation of training data caused by the combination of these methods, making the newly generated samples more challenging to identify and impacting the model's ability to learn useful features, leading to performance degradation. When multiple label-variant methods are combined, manifold intrusion

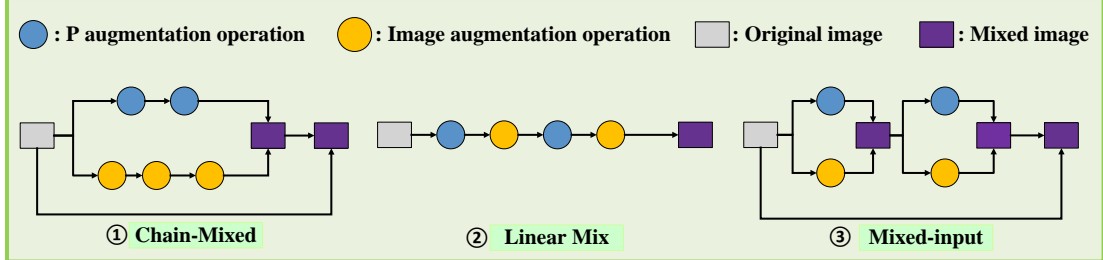

Figure 4: Different mixing framework of IPMix. P augmentation operation represents pixel-level and patch-level augmentation operations. ① Utilizing P operations and image-level operations in different chains and mixing the results. ② A clean image is randomly carried out by P operations or image-level operations in linear combinations to generate an IPMix image. ③ leveraging the mixed image as a new input.

issues may be more likely to arise. One possible solution for better information integration is to incorporate approaches (*e.g.*, MixUp) into search-based data augmentation techniques [20, 36]. However, searching the space for an optimal DA policy will bring expensive computation. Furthermore, this approach aims at improving clean accuracy and does not consider the overall safety performance.

## 4   IPMix: A Simple Method for Training Robust Classifiers

In this section, we propose IPMix, which integrates three levels of data augmentation methods into a label-preserving approach, comprehensively improving safety metrics without sacrificing clean accuracy. We first demonstrate how to merge various techniques into a coherent framework and then propose novel approaches to achieve superior information fusion.

### 4.1   Integrates Different Levels into A Coherent Approach

**Pixel-level & Patch-level.** As a label-preserving data augmentation approach, IPMix uses the equation below to mix two input images:

$$\tilde{x} = B \odot x_1 + (I - B) \odot x_2 \tag{1}$$

Where $x_1$ is the input image and $x_2$ represents an unlabeled synthetic image (*e.g.*, fractals, spectrum, or auto-generated contours). $B$ is a mask matrix suitable for both patch-level and pixel-level data augmentation methods, and $I$ is a binary mask filled with ones, having the same dimensions as $B$. $\odot$ represents the element-wise product. When performing mixing operations at the patch level, we choose a patch of random size and position from $B$, with a value of $\lambda$ (sample from Beta distribution) in this range and a value of 1 in other areas, which ensures that except for the mixing patch, the rest of the generated image comes from $x_1$. When performing mixing operations at the pixel level, we treat the entire image as a patch, with a value of $\lambda$. To make it efficient, we adopt fractals as representatives of synthetic data. However, IPMix is insensitive to mixing sets change, as shown in Table 8.

Fractals are geometric shapes with structural complexities and natural geometries. While previous works [32, 61] merely use iterated function systems (IFS) to create fractal data, we employ the Escape-time Algorithm for generating "orbit trap" complex fractals to enhance dataset complexity and diversity. Please refer to Appendix E for details about generating fractal images.

The above-described method provides two key advantages: (1) We utilize a simple approach to combine operations of two levels, facilitating better information fusion. (2) Our method is label-preserving, ensuring it is not affected by manifold intrusion while eliminating the need for label smoothing [62]. In the following sections, we refer to the method used in Eq. (1) as **P-level** data augmentation, signifying the employment of both patch-level and pixel-level methods.

**Image-level.** IPMix leverages various augmentation techniques and compositions to create a new image that does not deviate significantly from the original. Drawing inspiration from previous works [36, 37], we randomly sample operations from PIL (*e.g.*, brightness, sharpness) and randomly sample strengths to enhance the diversity of training data without expensive searching. Notably, these operations are disjoint from ImageNet-C corruptions, ensuring the robustness test's validity.

**The IPMix framework.** To determine the most effective methods for combining P-level and image-level, we conducted experiments using different mixing structures to generate a diverse set of IPMix images, as illustrated in the Figure 4 and Table 2. While Linear Mix achieves excellent results in clean accuracy and corruption robustness, it performs poorly in calibrated prediction confidence.

Table 2: Results are reported on CIFAR-100 and CIFAR-100-C with ResNeXt-29. The Chain-Mixed achieves the most balanced result on these metrics. Bold is best.

|  | Classification Error($\downarrow$) | Robustness mCE($\downarrow$) | Calibration RMS($\downarrow$) |
|---|---|---|---|
| **Chain-Mixed** | 18.3 | 28.1 | 3.8 |
| Linear Mix | **18.2** | **27.4** | 13.5 |
| Mixed Input | 19.8 | 29.6 | **3.6** |

Mixed Input performs better in calibration but is inferior in accuracy and corruption robustness compared to Chain-Mixed. Consequently, we chose Chain-Mixed as the default framework for IPMix. Furthermore, the experimental results highlight the potential of establishing a general framework for integrating various data augmentation methods.

## 4.2 Multi-scale Information Fusion

IPMix can enhance the diversity and the structural complexity of training data to improve model performance. However, we found that simple mixing methods restrict the model's capabilities. To overcome this issue, we use random mixing and scar-like image patches for achieving more effective information fusion.

**Random mixing.** In previous data augmentation works, it is typical to either linearly mix two images or extract specific image features, such as saliency [22, 42], which requires additional computations, for image mixing. As IPMix incorporates various levels of operations, its objective is to enhance the mixing of images, ultimately increasing data diversity. To accomplish this objective, IPMix employs four mixing operations: addition, multiplication, random pixel mixing, and random element mixing [63]. Random pixel mixing creates a binary mask of size $H \times W \times 1$ that operates on each channel sequentially, while random element mixing generates a binary mask of size $H \times W \times 3$ (RGB) that applies to all channels simultaneously. An example is shown in Figure 5. The experiments in Appendix B.1 show that both operations are beneficial to better information mixing between images and fractals.

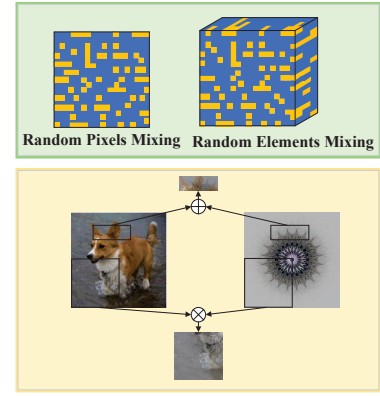

Figure 5: Top: Examples of random mixing operations. Bottom: Examples of IPMix-Scar mixing and IPMix-Square mixing.

**Scar-like image patches.** IPMix-Scar employs a long, thin rectangular box filled with an image patch to enhance dataset diversity, which has proven effective for anomaly detection [64]. An example of patch mixing is illustrated in Figure 5. First, IPMix randomly selects a point and a scar or square of the previously chosen size from the current image. Next, IPMix crops corresponding portions of the current image and the fractal picture and combine them.

Finally, we obtain IPMix, which employs various levels of data augmentation to create diverse transformations with image structural complexity and data diversity. Figure 3 displays an example of IPMix, where *k* denotes the number of augmented chains, and *t* represents the maximum number of times an image can be augmented. The algorithm of IPMix is summarized in Appendix D.

## 5 Experiments

In this section, we showcase the significant performance improvements brought by IPMix on clean datasets in multiple settings. We present the evaluation results of IPMix for image classification on three datasets—CIFAR-10, CIFAR-100 [65], and ImageNet [66]—across various models. Besides clean Classification, we assess IPMix on diverse safety tasks, including adversarial attack robustness, corruption robustness, prediction consistency, calibration, and anomaly detection. Please refer to Appendix C for details about the evaluation metrics. Lastly, we evaluate the properties of IPMix in thorough ablation studies and compare our approach with different levels of methods.

Table 3: Clean Error for IPMix on CIFAR-10 and CIFAR-100, lower is better. Top : CIFAR-10. Bottom : CIFAR-100. Mean and standard derivation over three random seeds is shown for each experiment. Bold is best.

| | Vanilla | MixUp | CutOut | CutMix | AugMix | PixMix | IPMix |
|---|---|---|---|---|---|---|---|
| WideResNet40-4 | $4.4_{(\pm 0.05)}$ | $3.8_{(\pm 0.06)}$ | $\mathbf{3.6}_{(\pm 0.05)}$ | $4.0_{(\pm 0.04)}$ | $4.3_{(\pm 0.08)}$ | $4.1_{(\pm 0.08)}$ | $4.0_{(\pm 0.06)}$ |
| WideResNet28-10 | $3.8_{(\pm 0.07)}$ | $3.6_{(\pm 0.08)}$ | $3.4_{(\pm 0.06)}$ | $3.4_{(\pm 0.05)}$ | $3.4_{(\pm 0.07)}$ | $3.8_{(\pm 0.13)}$ | $\mathbf{3.3}_{(\pm 0.08)}$ |
| ResNeXt-29 | $4.3_{(\pm 0.04)}$ | $\mathbf{3.8}_{(\pm 0.11)}$ | $4.2_{(\pm 0.08)}$ | $\mathbf{3.8}_{(\pm 0.02)}$ | $4.2_{(\pm 0.05)}$ | $\mathbf{3.8}_{(\pm 0.09)}$ | $\mathbf{3.8}_{(\pm 0.07)}$ |
| ResNet-18 | $4.4_{(\pm 0.05)}$ | $4.2_{(\pm 0.04)}$ | $4.1_{(\pm 0.05)}$ | $\mathbf{4.0}_{(\pm 0.04)}$ | $4.5_{(\pm 0.03)}$ | $4.4_{(\pm 0.05)}$ | $4.2_{(\pm 0.07)}$ |
| Mean | 4.2 | 3.9 | **3.8** | **3.8** | 4.1 | 4.0 | **3.8** |
| WideResNet40-4 | $21.3_{(\pm 0.11)}$ | $20.5_{(\pm 0.13)}$ | $19.9_{(\pm 0.11)}$ | $20.3_{(\pm 0.15)}$ | $20.6_{(\pm 0.15)}$ | $20.4_{(\pm 0.17)}$ | $\mathbf{19.4}_{(\pm 0.14)}$ |
| WideResNet28-10 | $19.0_{(\pm 0.13)}$ | $18.4_{(\pm 0.12)}$ | $18.8_{(\pm 0.15)}$ | $18.0_{(\pm 0.11)}$ | $19.4_{(\pm 0.11)}$ | $18.3_{(\pm 0.13)}$ | $\mathbf{17.4}_{(\pm 0.25)}$ |
| ResNeXt-29 | $20.4_{(\pm 0.11)}$ | $20.3_{(\pm 0.12)}$ | $19.6_{(\pm 0.13)}$ | $19.5_{(\pm 0.13)}$ | $20.4_{(\pm 0.13)}$ | $20.1_{(\pm 0.11)}$ | $\mathbf{18.3}_{(\pm 0.22)}$ |
| ResNet-18 | $23.7_{(\pm 0.09)}$ | $21.0_{(\pm 0.07)}$ | $22.0_{(\pm 0.11)}$ | $\mathbf{20.8}_{(\pm 0.12)}$ | $23.0_{(\pm 0.14)}$ | $21.6_{(\pm 0.15)}$ | $21.6_{(\pm 0.23)}$ |
| Mean | 21.1 | 20.0 | 20.1 | 19.7 | 20.8 | 20.1 | **19.2** |

Table 4: Mean Corruption Error (mCE) for IPMix across architectures on CIFAR-10-C and CIFAR-100-C, lower is better. Top : CIFAR-10-C. Bottom : CIFAR-100-C. Bold is best.

| | Vanilla | MixUp | CutOut | CutMix | AugMix | PixMix | IPMix |
|---|---|---|---|---|---|---|---|
| WideResNet40-4 | $26.4_{(\pm 0.14)}$ | $21_{(\pm 0.15)}$ | $25.9_{(\pm 0.13)}$ | $26_{(\pm 0.13)}$ | $10_{(\pm 0.12)}$ | $9.5_{(\pm 0.14)}$ | $\mathbf{8.6}_{(\pm 0.14)}$ |
| WideResNet28-10 | $24.2_{(\pm 0.15)}$ | $19.2_{(\pm 0.17)}$ | $23.5_{(\pm 0.17)}$ | $25.1_{(\pm 0.13)}$ | $9.1_{(\pm 0.14)}$ | $8.7_{(\pm 0.14)}$ | $\mathbf{7.5}_{(\pm 0.17)}$ |
| ResNeXt-29 | $27.5_{(\pm 0.11)}$ | $23.6_{(\pm 0.18)}$ | $27.3_{(\pm 0.18)}$ | $28.5_{(\pm 0.18)}$ | $11.3_{(\pm 0.15)}$ | $9.2_{(\pm 0.12)}$ | $\mathbf{8.6}_{(\pm 0.19)}$ |
| ResNet-18 | $25_{(\pm 0.09)}$ | $20_{(\pm 0.15)}$ | $24.1_{(\pm 0.13)}$ | $24.7_{(\pm 0.19)}$ | $10.4_{(\pm 0.13)}$ | $9_{(\pm 0.11)}$ | $\mathbf{8.4}_{(\pm 0.17)}$ |
| Mean | 25.8 | 20.9 | 25.2 | 26 | 10 | 9.1 | **8.2** |
| WideResNet40-4 | $50_{(\pm 0.15)}$ | $45.9_{(\pm 0.19)}$ | $51.5_{(\pm 0.17)}$ | $50_{(\pm 0.19)}$ | $33.3_{(\pm 0.22)}$ | $31.1_{(\pm 0.19)}$ | $\mathbf{28.6}_{(\pm 0.15)}$ |
| WideResNet28-10 | $48.5_{(\pm 0.21)}$ | $44.2_{(\pm 0.18)}$ | $48.2_{(\pm 0.15)}$ | $48.6_{(\pm 0.21)}$ | $31.5_{(\pm 0.21)}$ | $28.3_{(\pm 0.21)}$ | $\mathbf{26.6}_{(\pm 0.29)}$ |
| ResNeXt-29 | $51.4_{(\pm 0.19)}$ | $47.9_{(\pm 0.21)}$ | $51_{(\pm 0.17)}$ | $52.4_{(\pm 0.22)}$ | $34.1_{(\pm 0.24)}$ | $30.6_{(\pm 0.23)}$ | $\mathbf{28.1}_{(\pm 0.31)}$ |
| ResNet-18 | $50_{(\pm 0.18)}$ | $45.5_{(\pm 0.21)}$ | $50.2_{(\pm 0.19)}$ | $50.8_{(\pm 0.24)}$ | $35_{(\pm 0.25)}$ | $31.4_{(\pm 0.28)}$ | $\mathbf{29.9}_{(\pm 0.29)}$ |
| Mean | 50 | 45.9 | 50.2 | 50.5 | 33.4 | 30.3 | **28.3** |

We evaluate IPMix on CIFAR-10-C, CIFAR-100-C, and ImageNet-C to measure its resistance to corruption data shifts. We test IPMix on CIFAR-10-P, CIFAR-100-P, and ImageNet-P to measure network prediction stability against minor perturbations. To thoroughly demonstrate our method's capabilities, we assess it on supplementary datasets, including ImageNet-R, ImageNet-O, and ImageNet-A. Experiments on these datasets validate our approach's robustness under real-world distribution shifts.

## 5.1 Evaluation on CIFAR

We experiment with different backbone architectures on CIFAR-10 and CIFAR-100, including 40-4 Wide ResNet [67], 28-10 Wide ResNet, ResNeXt-29 [68], and Resnet-18 [69]. We compare IPMix with various data augmentation methods, including CutOut, MixUp, CutMix, AugMix, and PixMix. Please refer to Appendix A for more details about the training configurations.

**Accuracy.** In Table 3, we demonstrate that IPMix improves standard accuracy across architectures. In comparison with other approaches, IPMix achieves the best or comparable accuracy, showing the improvement of safety measures is not at the cost of hurting clean accuracy.

**Corruption robustness.** Results show that IPMix substantially improves corruption robustness across architectures. Compared to AugMix on CIFAR-100-C, IPMix achieves **4.7%**(40-4) and **4.9%**(28-10) improvement on WideResNet, **6%** on ResNeXt, and **5.1%** on ResNet. Table 4 demonstrates that IPMix achieves state-of-the-art results on both CIFAR-10-C and CIFAR-100-C.

**Calibration.** We utilize RMS calibration error [70] to evaluate the empirical frequency of correctness. As depicted in Figure 6, IPMix surpasses other methods, achieving state-of-the-art results.

**Prediction consistency.** We leverage the mean flip rate (mFR) to evaluate prediction consistency on CIFAR-10-P and CIFAR-100-P [51]. IPMix achieves the lowest mFR, as shown in Figure 7.

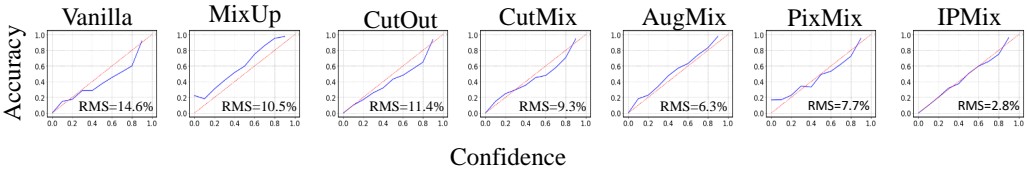

Figure 6: The results of calibration on CIFAR-100. IPMix achieves the lowest RMS error in all data augmentation methods, improving **11.8%** by comparing with Vanilla.

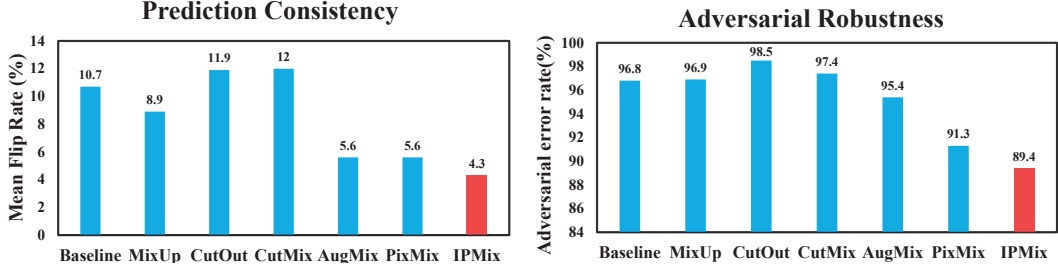

Figure 7: **Left: prediction consistency. Right: adversarial robustness.** IPMix achieves the best results on both metrics, demonstrating its ability to improve overall security performance.

**Adversarial robustness.** This measure evaluates the resistance of adversarially perturbed by projected gradient descent. We utilize PGD [71] to verify the adversarial robustness of image classifiers. The results in Figure 7 show that IPMix achieves the lowest error.

Table 5: The results of IPMix on ImageNet. For Anomaly Detection, we test the accuracy on ImageNet-A and AUPR on ImageNet-O, higher is better. IPMix achieves round improvement over various data augmentation methods. Bold is best, and underline is second.

| | Classification | Robustness | | Consistency | Calibration | | | Anomaly Detection | |
|---|---|---|---|---|---|---|---|---|---|
| | Clean Error(↓) | ImageNet-C mCE(↓) | ImageNet-R Error(↓) | ImageNet-P mFR(↓) | C RMS(↓) | R RMS(↓) | A RMS(↓) | ImageNet-A Classification(↑) | ImageNet-O AUPR(↑) |
| Vanilla | 23.9 | 78.6 | 64 | 57.7 | 12 | 19.9 | 47 | 2.2 | 16.2 |
| MixUp[23] | 22.7 | 76.5 | 62.4 | 54.6 | 9.3 | 41.7 | 49.3 | 5.2 | 16.1 |
| CutOut[28] | 22.6 | 73.1 | 64.6 | 57.9 | 11.3 | 19.7 | 46.3 | 4.7 | 15.9 |
| CutMix[40] | 22.9 | 77.2 | 66.5 | 58.1 | 9.6 | 44.2 | 48 | **7.2** | 16.5 |
| AugMix[37] | 22.6 | 68.5 | 61.8 | 52.3 | 8.1 | 13.1 | 43.5 | 3.8 | 17.4 |
| AugMax[29] | 22.9 | 67.4 | 62.1 | 54.6 | 8.8 | 12.1 | 44.7 | 3.9 | 17.1 |
| PixMix[24] | 22.4 | 65.4 | 59.8 | 50.8 | 7.2 | 12.3 | 44 | 5.9 | 17.3 |
| IPMix | **22.2** | **63** | **57.4** | **48.5** | **7.1** | **7** | **30** | 6.6 | **18.2** |

## 5.2 Evaluation on ImageNet

For ImageNet experiments, we compare different data augmentation methods, including MixUp, CutOut, CutMix, AugMix, AugMax [29], and PixMix. We utilize SGD optimizer with an initial learning rate of 0.01 to train ResNet-50 for 180 epochs following a cosine decay schedule. Please refer to Appendix A for more details about the training configurations.

IPMix achieves state-of-the-art or comparable performances on a broad range of safety measures, as shown in Table 5. Compared with other methods, IPMix improves the resistance of out-of-distribution shifts without reducing clean accuracy. On corruption robustness, IPMix outperforms Vanilla by **15.6%** and AugMix by **5.5%**, achieving state-of-the-art results. On ImageNet-R, IPMix demonstrates the ability to improve rendition robustness, increasing by **6.6%** by comparison with Vanilla. On ImageNet-P, IPMix improves mFR by **9.2%** over Vanilla and **2.3%** over PixMix. On calibration tests, IPMix surpasses all methods on ImageNet-C, ImageNet-R, and ImageNet-A, improving RMS by **0.1%**, **5.1%**, and **13.5%** by comparison with the second-best approach. Furthermore, IPMix achieves convincing results on ImageNet-A and ImageNet-O, demonstrating its exceptional ability in anomaly detection. The results demonstrate that IPMix can roundly improve safety metrics.

Table 6: Ablation results of different components of IPMix on ImageNet with ResNet-50.

| | Classification | Robustness | | Consistency | Calibration | | | Anomaly Detection | |
|---|---|---|---|---|---|---|---|---|---|
| | Clean Error($\downarrow$) | ImageNet-C mCE($\downarrow$) | ImageNet-R Error($\downarrow$) | ImageNet-P mFR($\downarrow$) | C RMS($\downarrow$) | R RMS($\downarrow$) | A RMS($\downarrow$) | ImageNet-A Classification($\uparrow$) | ImageNet-O AUPR($\uparrow$) |
| IPMix | **22.2** | **63** | **57.4** | **48.5** | **7.1** | **7** | **30** | **6.6** | **18.2** |
| w/o patch | 22.8$_{(\pm0.11)}$ | 65.1$_{(\pm0.16)}$ | 58.8$_{(\pm0.11)}$ | 49.1$_{(\pm0.15)}$ | 7.8$_{(\pm0.11)}$ | 7.4$_{(\pm0.08)}$ | 31.1$_{(\pm0.01)}$ | 6$_{(\pm0.01)}$ | 17.7$_{(\pm0.02)}$ |
| w/o pixel | 23.1$_{(\pm0.15)}$ | 65.6$_{(\pm0.19)}$ | 59.3$_{(\pm0.13)}$ | 49.5$_{(\pm0.17)}$ | 8.2$_{(\pm0.13)}$ | 7.4$_{(\pm0.09)}$ | 32.4$_{(\pm0.11)}$ | 5.6$_{(\pm0.01)}$ | 17.2$_{(\pm0.03)}$ |
| w/o image | 23.5$_{(\pm0.16)}$ | 66.2$_{(\pm0.21)}$ | 59.5$_{(\pm0.17)}$ | 49.6$_{(\pm0.14)}$ | 8.8$_{(\pm0.13)}$ | 8.1$_{(\pm0.13)}$ | 33.5$_{(\pm0.13)}$ | 6.5$_{(\pm0.02)}$ | 17.8$_{(\pm0.03)}$ |

## 5.3 Ablation Study

In this paragraph, we evaluate the properties of our approach by ablation experiments. We first study the influence of different parts of IPMix on performance and then assess the stability of IPMix under various mixing sources. Please refer to more ablation experiments in Appendix B.1.

**Components of IPMix.** In this section, we evaluate the influence of different IPMix components on performance.

Table 7: Ablation results of different components of IPMix on CIFAR-100. Mean and standard derivation over three random seeds is shown for each experiment. Bold is the best.

| | Classification | Robustness | Calibration |
|---|---|---|---|
| IPMix | **19.4** | **28.6** | **2.8** |
| w/o patch | 19.7$_{(\pm0.13)}$ | 30 $_{(\pm0.21)}$ | 4.6 $_{(\pm0.07)}$ |
| w/o pixel | 19.6 $_{(\pm0.09)}$ | 33 $_{(\pm0.35)}$ | 8.2 $_{(\pm0.12)}$ |
| w/o image | 20.1 $_{(\pm0.27)}$ | 34 $_{(\pm0.65)}$ | 8.6 $_{(\pm0.21)}$ |

We execute ablation experiments on the three primary IPMix constituents: image-level, patch-level, and pixel-level. The results show the indispensable contribution of each component to enhancing model performance, demonstrating that these approaches are complementary and that a unification among them is necessary to achieve robustness. The ablation experiment results are shown in Table 6 and Table 7. Please refer to thorough analysis in Appendix J.

**Mixing sources.** The excellent performance of IPMix is partly due to the structural complexity of fractal pictures. In this part, we examine the sensitivity of IPMix to different fractal sources on CIFAR-100. We report clean accuracy, corruption robustness, and calibration from different sources with WRN40-4. Fractal + FVis is the default setting of PixMix, which consists of fractals and feature visualization. FractalDB [59] consists of fractal images generated by Iterated Function System (IFS).

Table 8: Ablation results on IPMix across different mixing sets. The results show that IPMix is insensitive to mixing sets change.

| Mixing sets | Classification Error($\downarrow$) | Robustness mCE($\downarrow$) | Calibration RMS($\downarrow$) |
|---|---|---|---|
| Fractal + FVis | **19.4** | 28.8 | 3.3 |
| FractalDB | 20 | 29 | 5.4 |
| RCDB | 19.5 | **28.4** | 3.2 |
| Dead Leaves | **19.4** | 29.1 | 3.1 |
| Spectrum | 19.8 | 29.2 | 4 |
| fractals(ours) | **19.4** | 28.6 | **2.8** |

RCDB [60] consists of auto-generated contours. Dead Leaves and Spectrum generated from generative image models [31]. The full results show in Table 8.

## 5.4 The Comparison with Different Levels of Method

In this section, we perform an extensive performance comparison between IPMix and a range of existing methods using multiple metrics. We consider AutoAugment, RandAugment, and TrivialAugment [36] as representative image-level techniques, while SaliencyMix, PuzzleMix [72], and Co-Mixup [22] serve as typical patch-level techniques. For pixel-level methods, Manifold Mixup stands as our representative choice. IPMix does not require searching for the optimal DA policy like image-level techniques. In contrast to patch-level approaches, IPMix eliminates the need for saliency computations. The results in Table 9 show that IPMix outperformed all other methods on all metrics.

## 6 Analysis of IPMix

IPMix combines three levels of data augmentation into a unified, label-preserving technique to improve model performance. We believe that IPMix's superior performance is due to the increased

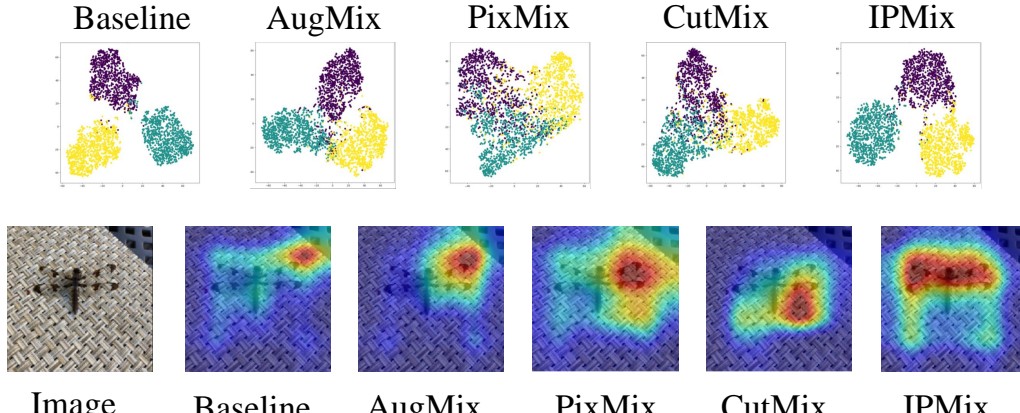

Figure 8: **Top: t-SNE visualization.** The features are from the penultimate layer of a WRN40-4 trained on CIFAR10. Compared with other approaches, IPMix has distinct boundaries between different category clusters and generates diverse samples to cover boundary areas, thereby improving the generalization ability. **Bottom: The Grad-CAM visualization**, with input images sourced from ImageNet-A, demonstrates that IPMix excels in identifying objects within complex scenarios.

Table 9: Results of different augmentation methods on CIFAR-100 and CIFAR-100-C with 28-10 Wide ResNet. Bold is best.

| Methods | Classification Error($\downarrow$) | Robustness mCE($\downarrow$) | Adversaries Error($\downarrow$) | Consistency mFR($\downarrow$) | Calibration RMS($\downarrow$) |
|---|---|---|---|---|---|
| AutoAugment [19] | $17.7_{(\pm0.11)}$ | $38.4_{(\pm0.15)}$ | $97.8_{(\pm0.22)}$ | $8_{(\pm0.06)}$ | $7.9_{(\pm0.06)}$ |
| RandAugment [20] | $17.8_{(\pm0.14)}$ | $41.5_{(\pm0.13)}$ | $96.6_{(\pm0.25)}$ | $8.6_{(\pm0.10)}$ | $7.9_{(\pm0.04)}$ |
| TrivialAugment [36] | $17.9_{(\pm0.13)}$ | $96.3_{(\pm0.21)}$ | $35.4_{(\pm0.23)}$ | $7.3_{(\pm0.07)}$ | $8.7_{(\pm0.04)}$ |
| SaliencyMix [42] | $18.3_{(\pm0.14)}$ | $38.3_{(\pm0.24)}$ | $96.7_{(\pm0.21)}$ | $10.8_{(\pm0.07)}$ | $7.1_{(\pm0.07)}$ |
| PuzzleMix [72] | $18.1_{(\pm0.11)}$ | $37.9_{(\pm0.21)}$ | $96.1_{(\pm0.23)}$ | $10.5_{(\pm0.04)}$ | $7.5_{(\pm0.08)}$ |
| Co-Mixup [22] | $18.0_{(\pm0.19)}$ | $35.6_{(\pm0.25)}$ | $95.6_{(\pm0.21)}$ | $10.1_{(\pm0.05)}$ | $7.7_{(\pm0.04)}$ |
| Manifold Mixup [39] | $18.8_{(\pm0.21)}$ | $51.3_{(\pm0.23)}$ | $93.4_{(\pm0.17)}$ | $29.9_{(\pm0.28)}$ | $10.2_{(\pm0.09)}$ |
| IPMix | $\mathbf{17.4}_{(\pm0.25)}$ | $\mathbf{26.6}_{(\pm0.29)}$ | $\mathbf{91.3}_{(\pm0.21)}$ | $\mathbf{4.2}_{(\pm0.11)}$ | $\mathbf{6.4}_{(\pm0.07)}$ |

data diversity and enhanced regularization effect. For a more intuitive demonstration of these effects, we utilize t-SNE and Class Activation Mapping (CAM) [73] for visualizations, as shown in Figure 8.

**Increasing diversity.** IPMix increases the diversity of training data by mixing data at multiple levels, enabling the model to learn a greater variety of feature combinations and patterns. Furthermore, the integration of synthetic data from distinct distributions (*e.g.*, fractals), further amplifies this diversity.

**Enhanced regularization effect.** The approach of mixing data also serves as a potent regularization technique. By randomly mixing samples, the model is compelled to learn more robust features rather than overly relying on specific sample or class characteristics, which reduces the risk of overfitting and enhances the model's performance in different environments.

## 7 Conclusion

We propose IPMix, which leverages different levels of augmentation techniques and image structural complexity to improve model performance. By employing random mixing methods, we facilitate more effective information fusion. The experimental results indicate that IPMix can significantly improve various safety metrics. We hope our work will attract attention to joining different methods into coherent and synergetic approaches to improve robustness and other safety measures. This adaptation is crucial given the growing importance of safety requirements in systems design.

## Acknowledgement

This work was under the help of Xi Yang. We thank him for his selfless help and valuable suggestions.

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
