**Contents of the Appendices:**

Section A. Details of experimental settings, hyperparameters, and configurations used in this paper.

Section B. Additional experiments of IPMix, including more ablation experiments, comparisons with other methods, and additional robustness experiments.

# A    Experimental Settings

## A.1    CIFAR

In this section, we share the training settings of IPMix on CIFAR. We experiment with various backbone architectures on CIFAR-10 and CIFAR-100, including 40-4 Wide ResNet, 28-10 Wide ResNet [1], ResNeXt-29 [2], and Resnet-18 [3]. We train ResNet and RexNeXt for 200 epochs, and all Wide ResNets for 100 epochs. We employ the SGD optimizer with a weight decay of 0.0001 and a momentum of 0.9. We randomly crop training images to $32\times32$ resolution with zero padding and flip them horizontally. We compare IPMix with various data augmentation methods, including CutOut, MixUp, CutMix, AugMix, and PixMix. We select a CutOut size of $16\times16$ pixels on CIAFR-10, and $8\times8$ on CIFAR-100. For CutMix, we set CutMix probability as 0.5 and $\alpha = 1.0$. We set $k = 3$ in AugMix, and $k = 3$, $\beta = 4$ in PixMix for the best results. For IPMix, we set $k = 3$, $t = 3$, and randomly select patch sizes from 4, 8, 16, and 32 (pixel-level). All experiments are conducted on a server with two NVIDIA GeForce RTX 3090 GPUs.

## A.2    ImageNet-1K

For ImageNet experiments, we compare different data augmentation methods, including MixUp, CutOut, CutMix, AugMix, AugMax [4], and PixMix. Since regularization methods may require a greater number of training epochs to converge, we fine-tune a pre-trained ResNet-50 for 180 epochs. We utilize SGD optimizer with an initial learning rate of 0.01 following a cosine decay schedule, with a batch size of 256. For all approaches, we randomly crop training images to $224\times224$ resolution with zero padding and flip them horizontally. We adopt $\alpha = 0.2$ for MixUp and CutMix and select a CutOut size of $56\times56$ pixels. For IPMix, we use $k = 3$, $t = 3$, and randomly select patch sizes from 4, 8, 16, 32, 64, and 256 (pixel-level). We set $\lambda = 12$ and $n = 5$ for AugMax-DuBIN, the same as the paper.

# B  Additional Experiments of IPMix

## B.1  Ablation Exmperiments

**IPMix hyperparameters.** In this paragraph, we evaluate the hyperparameters sensitivity of IPMix. We examine two hyperparameters: the number of augmented chains $k$ and the maximum image augmentation times $t$ with clean accuracy and robustness. The results in Table 1 demonstrate that IPMix is not sensitive to hyperparameters, showing the performance of IPMix is stable under change.

**Mixing operations ablation.** In this paragraph, we test IPMix's mixing operation sensitivity. IPMix utilizes four different operations to improve model performance, including addition, multiplication, random pixels mixing, and random elements mixing. The results show in Table 2.

**Patch mixing ablation.** In this paragraph, we verify IPMix's patch variants, which can be divided into two categories, IPMix-Scar and IPMix-Square. The results in Table 3 show that PachtMix-Scar can improve model robustness.

Table 1: We evaluate clean accuracy on CIFAR-100 and Mean Corruption Error (mCE) on CIFAR-100-C with WRN40-4. The performance of IPMix is not strongly associated with hyperparmeters.

|         | $k = 2$ | $k = 3$ | $k = 4$ |
|---------|---------|---------|---------|
| $t = 2$ | 19.5    | 19.3    | 19.3    |
|         | 29      | 28.9    | 29      |
| $t = 3$ | 19.7    | **19.4**| 19.7    |
|         | **28.5**| 28.6    | 28.6    |

Table 2: Ablation results of IPMix on CIFAR-100 with WRN40-4. While the addition + multiplication achieves the highest accuracy, it compromises corruption and calibration. In contrast, random mixing operations bolster robustness and calibration. Experiment results demonstrate that combining all mixing operations achieves the most balanced performance.

| Mixing operations       | Classification Error($\downarrow$) | Robustness mCE($\downarrow$) | Calibration RMS($\downarrow$) |
|-------------------------|:----------------:|:----------:|:-----------:|
| Addition + Multiplication | **19.2**       | 31         | 4.1         |
| Random pixels mixing    | 19.6             | 28.7       | 3.7         |
| Random elements mixing  | 19.9             | 28.8       | **2.7**     |
| IPMix                   | 19.4             | **28.6**   | 2.8         |

Table 3: The results of patch variants ablation on CIFAR-100 with ResNeXt-29.

| Variants     | Classification Error($\downarrow$) | Robustness mCE($\downarrow$) | Calibration RMS($\downarrow$) |
|--------------|:----------------:|:----------:|:-----------:|
| IPMix-Square | **18.3**         | 28.5       | 3.9         |
| IPMix-Scar   | 18.6             | **28.0**   | 4.1         |
| IPMix        | **18.3**         | 28.1       | **3.8**     |

## B.2  Additional Robustness Experiments

Recent works propose that some data augmentation techniques are tailored to particular datasets when testing model robustness. To evaluate the generality of IPMix, we experiment with other types of distribution shifts beyond common corruptions. We examine IPMix on CIFAR-10-$\overline{\text{C}}$, CIAFR-100-$\overline{\text{C}}$, and ImageNet-$\overline{\text{C}}$ [5]. CIFAR-10-$\overline{\text{C}}$, CIFAR-100-$\overline{\text{C}}$, and ImageNet-$\overline{\text{C}}$ are similar to CIFAR-C and ImageNet-C but utilize a different set of corruptions.Results in Table 4 demonstrate that IPMix achieves SOTA or comparison results by comparing with other methods.

Table 4: Results of robustness resist other distribution shifts. Bold is best.

| Methods | CIFAR-10-$\overline{\text{C}}$ | CIFAR-100-$\overline{\text{C}}$ | ImageNet-$\overline{\text{C}}$ |
|---|---|---|---|
| Vanilla | 26.4 | 52 | 60.2 |
| MixUp [6] | 22.4 | 50 | 54.1 |
| CutOut [7] | 24.2 | 50.1 | 58.4 |
| CutMix [8] | 25.1 | 49.9 | 57.8 |
| AugMix [9] | 19.3 | 41 | 54.3 |
| PixMix [10] | 13.6 | 36.7 | **47.1** |
| IPMix | **13** | **36** | 47.9 |

To better assess the performance of IPMix against natural distribution shifts, we extended our evaluation to various ImageNet benchmarks. We test IPMix on ObjectNet [11], ImageNet-E [12], ImageNet-Sketch [13], ImageNet-V2 [14], and Stylized-ImageNet [15]. The results presented in Table 5 indicate that IPMix consistently outperforms under diverse data shifts, underscoring its capability to enhance model robustness.

Table 5: Results of IPMix against natural distribution shifts. Higher is better.

| | ObjectNet | ImageNet-E | ImageNet-Sketch | ImageNet-V2 | Stylized-ImageNet |
|---|---|---|---|---|---|
| Vanilla | 17.3 | 76.7 | 24.2 | 63.3 | 7.4 |
| MixUp [6] | 18.4 | 77.1 | 24.4 | 63.6 | 7.3 |
| CutOut [7] | 17.3 | 24.1 | 58.4 | 63.7 | 7.6 |
| CutMix [8] | 18.9 | 76.7 | 23.8 | 65.4 | 5.3 |
| AugMix [9] | 17.6 | 78.6 | 28.5 | 65.2 | 11.2 |
| PixMix [10] | 18.5 | 80 | 29.2 | **65.8** | 11.8 |
| IPMix | **19.3** | **80.9** | **31.1** | 65.6 | **12.2** |

## C   Evaluation Metrics

We evaluate various safety measures on CIFAR and ImageNet, including corruption robustness, calibration, adversarial robustness, consistency, and anomaly detection. Task evaluation metrics are shown below.

**Corruption robustness.** Following AugMix, we utilize the Mean Corruption Error (mCE) to test a model's resistance to corrupted data on CIFAR-10-C, CIFAR-100-C, and ImageNet-C. Mean Corruption Error is the mean error rate normalized by the corruption errors of a baseline model over 15 corruption types and 5 corruption severity. We train AlexNet [16] as the baseline for ImageNet experiments.

**Calibration.** The calibration task is to verify whether the predicted probability estimates are representative of the true correctness likelihood. We use RMS Calibration Error [17] as the metric, which can be computed as $\sqrt{\mathbb{E}_C[(\mathbb{P}(Y = \hat{Y}|C = c) - c)^2]}$, where $C$ is the classifier's confidence that its prediction $\hat{Y}$ is correct. Lower is better.

**Adversarial robustness.** We utilize PGD to verify the adversarial robustness of image classifiers. We use 20 steps of optimization and an $\ell_\infty$ budget of 2/255 on CIFAR-10 and CIFAR-100. The metric is the classifier error rate. Lower is better.

**Consistency.** Following AugMix, we verify perturbation consistency on CIFAR-10-P, CIFAR-100-P, and ImageNet-P. The metric is the mean flip rate (mFR), which can be tested through video frame predictions normalized by a baseline model matched by 10 different perturbation types. We choose AlexNet as the baseline model.

**Anomaly detection.** We utilize two challenging datasets, ImageNet-A and ImageNet-O to evaluate model robustness under out-of-distribution shifts. The main metric on ImageNet-A is accuracy, and

on ImageNet-O is the area under the precision-recall curve (AUPR). Higher is better. The anomaly score is the negative of the maximum softmax probabilities [18].

## D   The Algorithm of IPMix

The algorithm to generate IPMix images is summarized in Algorithm 1. The fractals we use are selected at random from the IPMix fractal set (for further details, please see Appendix E). On CIFAR, the patch sizes we employ are randomly chosen from a set including 4, 8, 16, and 32, whereas for ImageNet-1K, we opt for patch sizes from 4, 8, 16, 32, 64, and 256. We randomly mix the augmented original image to increase diversity. Across all our experiments, we consistently use $k = 3$ and $t = 3$.

---

**Algorithm 1:** Generate IPMix Images

---

**input** : Origin image $x$, fractal $x_{\text{fractal}}$, augmentation methods $M$={image-level, P-level}, patch sizes $P_{\text{size}}$ , P operations $P$ = {random pixels mixing,...,add}, image operations $I$ = {invert,...,mirror} , width $k$, max depth $t$.

**output** : $x_{\text{IPMix}}$

1 Sample mixing weights $w_1,...,w_k \sim$ Dirichlet$(\alpha,...,\alpha)$
2 Sample weights $m \sim$ Beta$(\alpha,\alpha)$
3 Generate $x_{\text{mix}}$ = Zerolikes$(x)$
4 **for** $i \leftarrow 1$ **to** $k$ **do**
5     Generate $x_{\text{mixed}} = x.\text{copy}()$
6     Randomly choose method 'meth' from $M$
7     **if** 'meth' == 'P-level' **then**
8         **for** $j = 1$ **to** random.choose([1,...,$t$]) **do**
9             Random sample size $s$ from $P_{\text{size}}$         `// Psize = x.size → Pixel-level op`
10             Sample operations $p_o$ from $P$
11             **if** random.random() $> 0.5$ **then**
12                 $x_{\text{mixed}}$ = patch mixing$(x_{\text{mixed}}, x_{\text{fractal}}, s, p_o)$       `// See Sec.4.2`
13             **else**
14                 Sample operations $i_o$ from $I$         `// For diversity increase`
15                 $x_{\text{aug}} = i_o(x)$
16                 $x_{\text{mixed}}$ = patch mixing$(x_{\text{mixed}}, x_{\text{aug}}, s, p_o)$
17     **else**
18         **for** $j = 1$ **to** random.choose([1,...,$t$]) **do**
19             Sample operations $i_o$ from $I$
20             $x_{\text{mixed}} = i_o(x_{\text{mixed}})$
21     $x_{\text{mix}}$ += $w_i \cdot x_{\text{mixed}}$         `// wi from Dirichlet(α,...,α)`
22 **return** $x_{\text{IPMix}} = m \cdot x_{\text{mix}} + (1 - m) \cdot x$         `// m from Beta(α,α)`

---

## E   Generating Fractal Images

While prior works have exclusively utilized Iterated Function Systems (IFS) to generate fractal data [19, 20], various other fractal-generating programs can also be employed. To further enhance the structural complexity and diversity, we have ventured beyond IFS and incorporated the Escape-time Algorithm to generate 'orbit trap' complex fractals. The most common 'orbit trap' fractal images, Mandelbrot and Julia fractals, can be derived from Eq. (1):

$$F(z) = z^2 + c \tag{1}$$

In Eq. (1), $z$ represents a complex number, and $c$ is a constant value. In the case of the Mandelbrot set, we initialize $z$ at 0, with $c$ corresponding to the specific coordinate in the complex plane that is under examination. Conversely, when generating the Julia set, $c$ remains constant throughout the set, and $z$ is initiated as the particular coordinate that is currently being tested.

Moreover, guided by the approach of [20], we create an additional 3000 fractals, each rendered with a unique, randomly generated background and color scheme using IFS. Furthermore, we supplement our dataset with an additional fractals obtained from DeviantArt[1]. These images, exhibiting greater

---

[1]`https://www.deviantart.com/`

complexity than those generated via IFS or the Escape-time Algorithm, significantly enhance dataset diversity. Besides, we collect 4000 feature images to improve diversity. In total, we assemble a collection of 13000 images named IPMix set for increasing data diversity and structural complexity when mixed with clean images.

## F   The Details about Combination Experiments

In this section, we show that simply combining different levels of approaches can degrade model performance across various metrics. Building upon these findings, in this part, we want to examine the impact of the order of operations on combination experiments.

In our experiments, we adopt MixUp [6], CutMix [8], and AugMix [9] as representative techniques for pixel-level, patch-level, and image-level augmentation, respectively. In all experiments, we apply AugMix first, followed by CutMix or MixUp. The rationale behind this order is that AugMix is commonly used in PIL images to enhance data diversity. In contrast, MixUp and CutMix interpolate and mix images after images conversion into tensors. Furthermore, applying Mixup/CutMix before AugMix could lead to unnatural transformations, as AugMix operations would distort the mixed images, counteracting the aim of preserving the individual image context during interpolation.

We have adopted several different combinations as follows.

- First, we apply AugMix, then MixUp, and finally CutMix.

- First, we apply AugMix, then CutMix, and finally MixUp.

- We apply AugMix first, followed by either CutMix or MixUp, chosen randomly.

- We apply AugMix first. Depending on the training epochs, we use either CutMix or MixUp.

Table 6: The combination experiments of different levels of data augmentation on CIFAR-100.

| Methods | Classification Error($\downarrow$) | Robustness mCE($\downarrow$) | Calibration RMS($\downarrow$) |
|---|---|---|---|
| Vanilla | 21.3 | 50 | 14.6 |
| MixUp | 20.5 | 45.9 | 10.5 |
| CutMix | 20.3 | 50 | 9.3 |
| AugMix | 20.6 | 33.3 | 6.3 |
| AugMix$\rightarrow$MixUp$\rightarrow$CutMix | 23.4 | 50.1 | 25.6 |
| AugMix$\rightarrow$CutMix$\rightarrow$MixUp | 27 | 51.4 | 26.7 |
| Chosen Randomly ($p = 0.5$) | 22.6 | 40.6 | 19 |
| Epoch-Dependent | 21.1 | 37.6 | 7.2 |

In all experiments, we use the optimal hyperparameters specified in the original papers. We set $k = 3$ for AugMix and $\alpha = 1$ for MixUp and CutMix. The results are demonstrated in Table 6.

We set the total number of training epochs to 100 on 40-4 Wide ResNet for all experiments. In our Epoch-Dependent combination experiments, we found that employing MixUp for the initial 50 epochs and transitioning to CutMix for the rest yielded the best performance. Nevertheless, it doesn't perform as well as the individual augmentation techniques. This underperformance might be due to the increased complexity in the synthesized training instances, possibly impeding the extraction of discriminative feature representations by models. Further experiments could explore different combinations of these techniques to improve their effectiveness.

In order to thoroughly analyze the influence of the augmentation strength of each method, we have conducted experiments considering various hyperparameter combinations. Specifically, we evaluated $k = 1, 3, 5$ (for AugMix) and $\alpha = 0.2, 0.5, 1$ (for MixUp and CutMix). We opted to exclude $k = 3$ and $\alpha = 1$, the original optimal hyperparameters in their papers, thereby reducing the total combinations from 27 to 8. From the experimental results in Table 7, combining different hyperparameters does not significantly improve the model performance. We set the total number of training epochs to 100 for all experiments with WRN40-4 on CIFAR-100.

Table 7: Could decreasing the augmentation strength of each method yield better performance?

| Combination | Classification Error($\downarrow$) | Robustness mCE($\downarrow$) | Calibration RMS($\downarrow$) |
|---|---|---|---|
| $\alpha = 0.2, \alpha = 0.2, k = 1$ | 23.9 | 51.2 | 25.3 |
| $\alpha = 0.2, \alpha = 0.2, k = 5$ | 24.5 | 51 | 25.3 |
| $\alpha = 0.2, \alpha = 0.5, k = 1$ | 26 | 50.7 | 24.9 |
| $\alpha = 0.2, \alpha = 0.5, k = 5$ | 24.4 | 50.6 | 25.7 |
| $\alpha = 0.5, \alpha = 0.2, k = 1$ | 25.8 | 50.8 | 25.4 |
| $\alpha = 0.5, \alpha = 0.2, k = 5$ | 25 | 49.1 | 24.8 |
| $\alpha = 0.5, \alpha = 0.5, k = 1$ | 25.5 | 50.5 | 25.1 |
| $\alpha = 0.5, \alpha = 0.5, k = 5$ | 26 | 51.2 | 25.9 |

## G   Training Time

In this section, we present a comparative analysis of the training time. The results in Table 8 show that IPMix adds only a modest training overhead over Vanilla, which is advantageous for its practical use in real-world scenarios.

Table 8: We test IPMix on two NVIDIA GeForce RTX 3090 GPUs with ResNet18 for 90 epochs. The training time of IPMix is acceptable by comparison with other data augmentation methods.

| Method | Time(sec/epochs) |
|---|---|
| Vanilla | 3764 |
| MixUp [6] | 3913 |
| CutOut [7] | 3870 |
| CutMix [8] | 4139 |
| AugMix [9] | 4762 |
| PixMix [10] | 4310 |
| AugMax [4] | 7564 |
| IPMix | 4380 |

## H   Full Results of IPMix across Architectures

In Table 9, we show the full results of IPMix across architectures on CIFAR-10 and CIFAR-100.

Table 9: Full results for IPMix on CIFAR-10 and CIFAR-100. We test the ability of IPMix on accuracy, robustness, consistency, adversaries, and calibration across different models. Top: CIFAR-10. Bottom : CIFAR-100.

| | Classification Error($\downarrow$) | Robustness mCE($\downarrow$) | Consistency mFR($\downarrow$) | Adversaries Error($\downarrow$) | Calibration RMS($\downarrow$) |
|---|---|---|---|---|---|
| WideResNet40-4 | 4 | 8.6 | 1.3 | 74.4 | 2.3 |
| WideResNet28-10 | 3.3 | 7.5 | 1.1 | 76.4 | 1.9 |
| ResNeXt-29 | 3.8 | 8.6 | 1.4 | 93.2 | 2 |
| ResNet-18 | 4.2 | 8.4 | 1.7 | 80 | 2.4 |
| Mean | 3.8 | 8.3 | 1.4 | 81 | 2.2 |
| WideResNet40-4 | 19.4 | 28.6 | 4.3 | 89.4 | 2.8 |
| WideResNet28-10 | 17.4 | 26.6 | 4.2 | 91.3 | 6.4 |
| ResNeXt-29 | 18.3 | 28.1 | 5 | 96.9 | 3.8 |
| ResNet-18 | 21.6 | 29.9 | 5.4 | 95.6 | 6.3 |
| Mean | 19.2 | 28.3 | 4.7 | 93.3 | 4.9 |

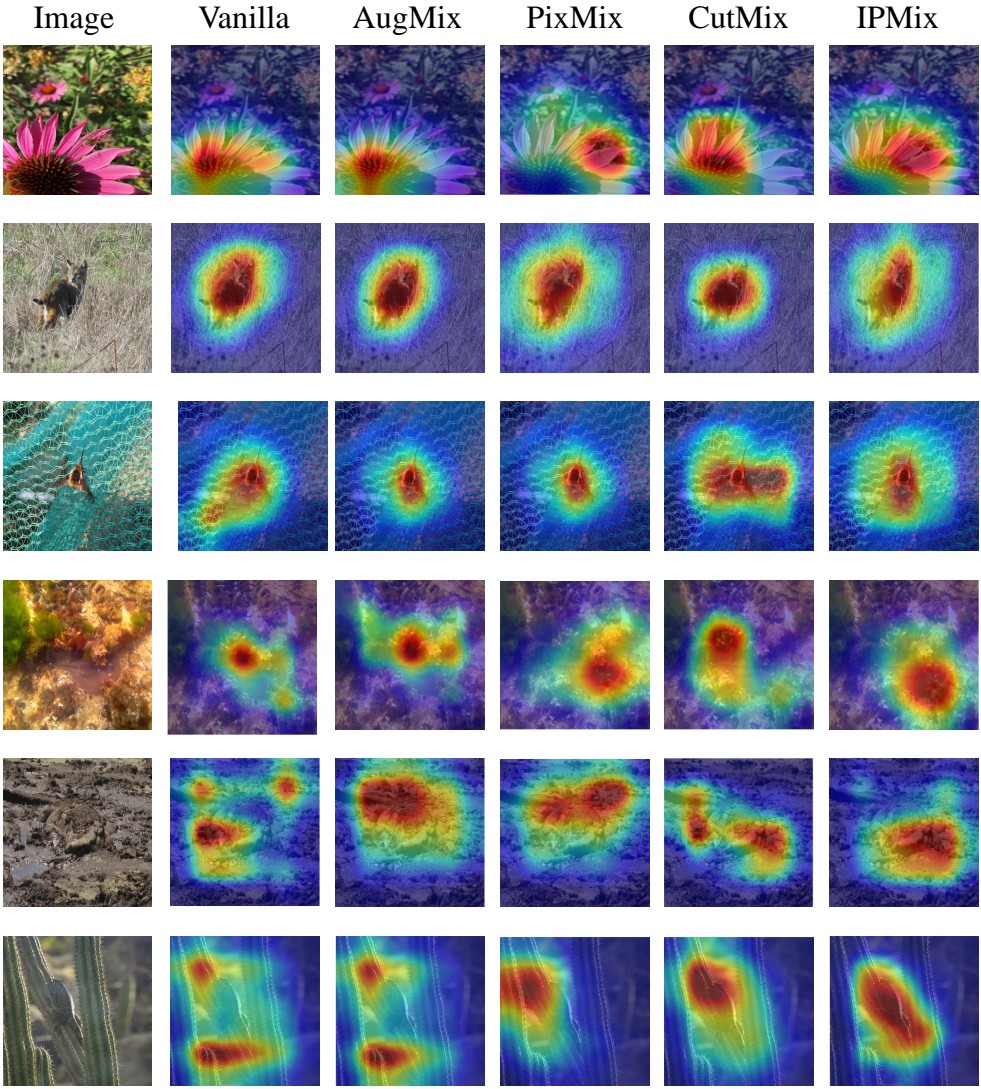

| Image | Vanilla | AugMix | PixMix | CutMix | IPMix |
|-------|---------|--------|--------|--------|-------|

Figure 1: More CAM visualizations of IPMix. Input images come from ImageNet-A, the most challenging dataset to verify the performance of model classifiers against distribution shifts.

## I More CAM Visualizations

In this section, we demonstrate more CAM visualizations of IPMix, as shown in Figure 1.

## J The Analysis of Ablation Experiments

In this section, we will detailed analyze the impact of each part on different safety metrics through ablation experiment results shown in Table 10.

**Accuracy**: The image-level augmentation has the most substantial effect on accuracy, aligning with current findings [21, 22] that image-level methods are commonly used to boost accuracy.

**Robustness**: Both pixel-level and image-level augmentations improve robustness. Since pixel-level introduces fine-grained variations for pattern recognition, while image-level increases dataset diversity, preventing the model from merely memorizing fixed augmentations.

Table 10: Ablation results of different components of IPMix on CIFAR-100. Mean and standard derivation over three random seeds is shown for each experiment. Bold is the best.

|  | Classification Error($\downarrow$) | Robustness mCE($\downarrow$) | Consistency mFR($\downarrow$) | Adversaries Error($\downarrow$) | Calibration RMS($\downarrow$) |
|---|---|---|---|---|---|
| IPMix | **19.4**$_{(\pm 0.17)}$ | **28.6**$_{(\pm 0.2)}$ | 89.4$_{(\pm 0.18)}$ | **4.3**$_{(\pm 0.09)}$ | **2.8**$_{(\pm 0.07)}$ |
| w/o patch | 19.7$_{(\pm 0.13)}$ | 30 $_{(\pm 0.21)}$ | 91.7 $_{(\pm 0.15)}$ | 4.7 $_{(\pm 0.02)}$ | 4.6 $_{(\pm 0.07)}$ |
| w/o pixel | 19.6 $_{(\pm 0.09)}$ | 33 $_{(\pm 0.35)}$ | 92.6 $_{(\pm 0.20)}$ | 5.2 $_{(\pm 0.05)}$ | 8.2 $_{(\pm 0.12)}$ |
| w/o image | 20.1 $_{(\pm 0.27)}$ | 34 $_{(\pm 0.65)}$ | **87.8** $_{(\pm 0.22)}$ | 5.5 $_{(\pm 0.11)}$ | 8.6 $_{(\pm 0.21)}$ |

**Calibration and Consistency**: The Image-level part significantly influences calibration and consistency, which increases diversity to improve the prediction calibration across scenarios and ensures consistency in responses to minor perturbations.

**Adversarial Attacks**: Without the image-level component, adversarial performance improves, implying diverse data might **weaken** defense against attacks. Conversely, removing pixel-level methods will degrade adversarial robustness, given their inherent resistance to perturbations.

## K  The Experiment Results on Transformer Architecture

In this section, we will evaluate the performance of IPMix on Vision Transformer. We trained a small ViT for 300 epochs on CIFAR-10 and CIFAR-100. This step aimed to confirm IPMix's potential on smaller datasets using Transformer architectures. In future work, we plan to expand our experiments with transformer architectures. The experiment results in Table 11 and Table 12 show that IPMix achieves the best performance on ViT.

Table 11: Experiments on CIFAR-10. Bold is the best.

|  | Classification Error($\downarrow$) | Robustness mCE($\downarrow$) | Consistency mFR($\downarrow$) | Adversaries Error($\downarrow$) | Calibration RMS($\downarrow$) |
|---|---|---|---|---|---|
| Vanilla | 19.5$_{(\pm 0.07)}$ | 27.7$_{(\pm 0.14)}$ | 91.3$_{(\pm 0.13)}$ | 5.9$_{(\pm 0.02)}$ | 10 |
| MixUp | 1$_{(\pm 0.11)}$ | 34.7$_{(\pm 0.21)}$ | 89.3$_{(\pm 0.21)}$ | 6$_{(\pm 0.05)}$ | 9.9$_{(\pm 0.03)}$ |
| CutMix | 19.3$_{(\pm 0.08)}$ | 34.3 $_{(\pm 0.19}$ | 89.1$_{(\pm 0.14)}$ | 5.5$_{(\pm 0.05)}$ | 7.5 $_{(\pm 0.02)}$ |
| PixMix | 28.4$_{(\pm 0.14)}$ | 33.$_{(\pm 0.24)}$ | 91$_{(\pm 0.12)}$ | 6.5$_{(\pm 0.11)}$ | 4.4$_{(\pm 0.07)}$ |
| AugMix | 20.3$_{(\pm 0.14)}$ | 25.6$_{(\pm 0.2)}$ | 80.3$_{(\pm 0.16)}$ | 5.1$_{(\pm 0.09)}$ | 6$_{(\pm 0.08)}$ |
| IPMix | **19.2**$_{(\pm 0.12)}$ | **23.7**$_{(\pm 0.2)}$ | **75.8**$_{(\pm 0.13)}$ | **3.7**$_{(\pm 0.07)}$ | **5.3**$_{(\pm 0.07)}$ |

Table 12: Experiments on CIFAR-100. Bold is the best.

|  | Classification Error($\downarrow$) | Robustness mCE($\downarrow$) | Consistency mFR($\downarrow$) | Adversaries Error($\downarrow$) | Calibration RMS($\downarrow$) |
|---|---|---|---|---|---|
| Vanilla | 40.1$_{(\pm 0.12)}$ | 56.3$_{(\pm 0.1)}$ | 96.2$_{(\pm 0.14)}$ | 12.4$_{(\pm 0.04)}$ | 14.8$_{(\pm 0.02)}$ |
| MixUp | 40$_{(\pm 0.14)}$ | 56 $_{(\pm 0.18)}$ | 92.5$_{(\pm 0.18)}$ | 9.8$_{(\pm 0.03)}$ | 9.5 $_{(\pm 0.02)}$ |
| CutMix | 39.5$_{(\pm 0.11)}$ | 56.3 $_{(\pm 0.15}$ | 96.2$_{(\pm 0.17)}$ | 10$_{(\pm 0.03)}$ | 9.8 $_{(\pm 0.03)}$ |
| PixMix | 48.7$_{(\pm 0.14)}$ | 54.3$_{(\pm 0.21)}$ | 93.2$_{(\pm 0.14)}$ | 10.9$_{(\pm 0.17)}$ | 4.9$_{(\pm 0.04)}$ |
| AugMix | 35.3$_{(\pm 0.17)}$ | 42.4$_{(\pm 0.21)}$ | 84.6$_{(\pm 0.16)}$ | 6.9$_{(\pm 0.03)}$ | 6.4$_{(\pm 0.07)}$ |
| IPMix | **32.6**$_{(\pm 0.11)}$ | **39.6**$_{(\pm 0.23)}$ | **83.2**$_{(\pm 0.15)}$ | **6.3**$_{(\pm 0.04)}$ | **5.3**$_{(\pm 0.05)}$ |

## L  The Drawbacks of Different Levels of Methods

In this section, we will reveal the drawbacks of different levels of approaches and explain how IPMix solves these problems.

The drawbacks of label variant methods:

**Pixel-level:** Mixing images with distinct labels and linearly interpolating between them will impose certain "local linearity" constraints on the model's input space beyond the data manifold, which may lead to "manifold intrusion". Consider one experiment on MNIST. If we use MixUp to linearly mix two numbers, such as "1" and "5", the generated image will show the characteristics of "8". When the generated "8" collides with a real "8" in the data manifold, there will be a problem of manifold intrusion. Since the two samples have similar characteristics, one is the real label and the other is a soft label ("1" and "5"). This will interfere with its ability to understand and classify categories and degrade model performance.

**Patch-level:** The problem of manifold intrusion also occurs in the patch-level method, termed "label mismatch." This occurs when the chosen source patch doesn't accurately represent the source object, leading the interpolated label misleads the model to learn unexpected feature representation. For example, using CutMix to mix images of a cat and a dog. CutMix might select 20 % of the background area from the cat image without information about the object (cat). However, their interpolated labels encourage the model to learn both objects' features (dog and cat) from that training image and degrade model performance.

The drawbacks of image-level methods:

**Image-level** data augmentation increases data diversity by applying label-preserving transformations to the whole image. Notable among these are search-based methods like AutoAugment, RandAugment, and FastAugment. While they improve performance effectively, the computationally expensive search for an optimal augmentation policy often exceeds the training process's complexity. Thus, efforts to minimize the search space, optimize search parameters, and uncover potential universal pipelines are central to the effectiveness of these methods.

In conclusion, we solve these questions by:

- Incorporate structural complexity from synthetic data at various levels to produce more diverse images. Our method is **label-preserving**, ensuring it is not affected by manifold intrusion.
- Randomly sample operations from PIL (e.g., brightness, sharpness) and randomly sample strengths to enhance the diversity of training data **without expensive searching**.
- Integrate three levels of data augmentation into a single framework with limited computational overhead, demonstrating that these approaches are complementary and that a unification among them is necessary to achieve robustness.

## M   Limitation and Broader Impact

While IPMix has shown promising results, the theoretical foundation of IPMix requires further development to gain deeper insights into its underlying principles. Meanwhile, our approach primarily focuses on CNN, and its effectiveness on Visual Transformers requires additional experimental validation. Additionally, the experiments conducted on a limited set of safety metrics, and the performance of IPMix in real-world scenarios with more comprehensive safety measures warrants future investigation [23]. In continuing our efforts to refine and enhance the IPMix methodology, we will focus on addressing these limitations in future works.

Since IPMix improves various safety measures, it can generate many beneficial effects in real-world environments, improving the robustness against attacks and the calibrated prediction confidence of models. Moreover, IPMix integrates three levels of data augmentation into a single framework, demonstrating that these approaches are complementary and necessary to achieve better performance. We believe the improvements in safety metrics and the coherent framework of combining various techniques will shed light on this field.