# OpenReview forum: "IPMix: Label-Preserving Data Augmentation Method for Training Robust Classifiers"
_NeurIPS.cc/2023/Conference — NeurIPS 2023 poster_

### Official Review · Reviewer_jACH · 2023-07-02

**Soundness:** 2 fair
**Presentation:** 3 good
**Contribution:** 2 fair
**Rating:** 5
**Confidence:** 5

**Summary:**

This paper proposes IPMix, a data augmentation strategy to improve model robustness (and other safety metrics like calibration). The idea builds upon PixMix, a recent data augmentation strategy generating diversity via synthetic pictures. IPMix performs this at multiple scales (pixel, patch, image levels) and mixes the resulting transformed images. The evaluations are performed on CIFAR-10, CIFAR-100, ImageNet and their shifted versions against several data augmentation baselines.

**Strengths:**

The paper is very well written, it is quite easy to understand. The method is clear and reasonable, given that distribution shifts may happen at different levels it makes sense to address them at different levels. The comparisons and ablation studies are well thought. Additional experimental results in supplementary are also helpful.


**Weaknesses:**

My main concerns are about ImageNet results which suggest minor improvements over PixMix, e.g. ImageNet-C results are 78.1 vs 77.8. The results are close for other benchmarks too like ImageNet-R and ImageNet-P (very close to AugMix). Same goes for anomaly detection results. This makes me wonder: Does using synthetic data and mixing have inherent limitations in terms of generating diversity? For example, what happens if you increase your diversity using recent diffusion models to generate data, e.g. as in [A]?

My other questions and comments:
- I would be interested in seeing comparisons to DeepAugment as well, as it has been shown to improve robustness notably when combined with AugMix.
- How does IPMix perform against natural distribution shifts like ImageNet-V2 and ObjectNet? What about more recent ImageNet benchmarks like ImageNet-3DCC and ImageNet-E? I believe having these evaluations could also provide some insights into the limitations I mentioned above.
- When I was reading L201 I thought about the work of Mintun et al. ([15] in supmat) and I was happy to see that you added evaluations on CIFAR-C-bar in the supplementary, which again seems to suggest minor improvements over PixMix. What would be the results for ImageNet-C-bar?
- It is nice that you added training overhead in supmat too, but I think it should be better linked in the main paper as I was going to ask about it and it wasn’t pointed out in the main paper.
- I understand this is a CNN based study but given the landscape and domination of ViTs I believe evaluations on ViT backbones are also necessary to truly understand the performance of IPMix.

[A] Sariyildiz et al. Fake It Till You Make It: Learning Transferable Representations From Synthetic ImageNet Clones, CVPR 2023.


**Questions:**

Please see weaknesses.


**Limitations:**

Please see weaknesses.

After rebuttal, I am increasing my rating to borderline accept.

---

> ### Author Rebuttal · Authors · 2023-08-09
>
> Thank you for your detailed concerns.
>
> W1: First and foremost, I'd like to emphasize that the improvements we achieved are not minor. Our approach consistently shows improvement across various safety metrics by comparison with other methods, which is more meaningful from a practical perspective.
>
> We value your suggestion. While employing diffusion to generate new samples on ImageNet indeed introduces diversity[A], it inevitably alters the labels during mixing with origin images (Since generated dogs are still dogs). Mixing directly with these generated images may violate our original intention of label-preserving.
>
> Inspired by your idea, we designed a feasible experiment that applied the diffusion model to our IPMix set. Specifically, we use stable diffusion to generate an additional 10000 images, leading to a combined dataset of 20000 images. Due to time constraints and limited computational resources during the rebuttal period, our experiments were only conducted on CIFAR-10/100 using the WRN28-10. Mean and standard derivation over three random seeds is shown for each experiment.
>
> **Experiments on CIFAR-100 with WRN28-10:**
>
> ||Classification (Error)↓|Robustness (mCE)↓ |Adversaries (Error)↓|Consistency (mFR)↓|Calibration (RMS)↓|
> |-|-|-|-|-|-|
> |IPMix|17.4 $ \pm $0.17|26.6 $ \pm $0.19|91.3 $ \pm $0.21|4.2 $ \pm $0.11|**6.4 $ \pm $0.07**|
> |IPMix+diffusion|**17.3 $ \pm $0.15**|**26 $ \pm $0.25**|**91.2 $ \pm $0.17**|**4.1 $ \pm $0.13**|6.5 $ \pm $0.09|
>
> **Experiments on CIFAR-10 with WRN28-10:**
>
> ||Classification (Error)↓|Robustness (mCE)↓ |Adversaries (Error)↓|Consistency (mFR)↓|Calibration (RMS)↓|
> |-|-|-|-|-|-|
> |IPMix|3.3 $ \pm $0.17|7.5 $ \pm $0.23|**76.4 $ \pm $0.23**|1.1 $ \pm $0.10|1.9 $ \pm $0.16|
> |IPMix+diffusion|**3.3 $ \pm $0.14**|**7.2 $ \pm $0.21**|76.5 $ \pm $0.19|**1.0 $ \pm $0.08**|**1.8 $ \pm $0.11**|
>
> From the experimental results, after using the diffusion model to generate new samples, the performance of IPMix has a minor improvement.
>
> From the current research and our experimental results, it seems premature to provide a definitive answer to "whether there are inherent limitations of synthetic data to generate diversity". Mixing data requires a thorough evaluation of the **original data distribution** and the **nature of synthetic data**. With the wide range of image generation methods available today, such as diffusion-based, statistical, and procedural image models, it's challenging to pinpoint *which type of synthetic data maximizes diversity enhancement for specific data distribution*. A potential solution might be to adopt an AutoAugment approach, using different synthetic data for various datasets, and conducting an extensive search through the augmentation policy space to identify the optimal synthetic data suited for a particular dataset. Nevertheless, this approach entails a significant computational cost, often exceeding the expense of training. Interestingly, our experiments suggest that **combining multiple types of synthetic data** (e.g., diffusion model and fractal data) seems to yield better results. This inspires us that there may be a way to mix different types of synthetic data to **maximize the diversity** of different data sets. We believe this could be a promising research direction and plan to explore it further in our future work.
>
> Q1: Thank you for your question.
>
> Due to time and computational constraints during the rebuttal, we were unable to train a model from scratch on ImageNet with DeepAugment+IPMix. As an alternative, we conducted our experiments on Tiny-ImageNet. Specifically, Tiny-ImageNet was chosen because it retains a data distribution similar to ImageNet but has a significantly reduced dataset size, making our experiments more manageable. On Tiny-ImageNet, we trained both AugMix, IPMix, DeepAugment+AugMix and DeepAugment+IPMix for 300 epochs on ResNet-18 and evaluated metrics on Tiny-ImageNet-C. Below are our experimental results:
>
> |Method|Accuracy(↑)|mCE(↓)
> |-|-|-|
> |Vanilla| 61.5|100|
> |AugMix| 61.8|83.2|
> |IPMix| 62.9|80.1|
> |DeepAugment+AugMix| 59.6|78.4|
> |DeepAugment+IPMix| **60.6**(+1)|**75.7**(-2.7)|
>
> From the experimental results, combining DeepAugment with IPMix significantly improves robustness. However, this improvement comes at the cost of loss of accuracy, demonstrating a trade-off between accuracy and robustness. It's possible that DeepAugment introduces distorted images that improve robustness, which might be misaligned with the origin data distribution, thereby compromising clean accuracy.
>
> Q2&Q3: Thanks for your questions.
>
> We have conducted experiments on different data sets. The experimental results are shown in the table below:
>
> **Experiments on ImageNet with ResNet-18:**
>
> |Method|ObjectNet|Imagenet-sketch|Imagenet-style|Imagenetv2|ImageNet-E|ImageNet-3DCC|ImageNet-C-bar|
> |-|-|-|-|-|-|-|-|
> |AugMix| 9.7|22.6|9.5|55.8|70.5|24.1|38.4|
> |PixMix| 10.8|22.1|**10.8**|54.9|69.7|**27.4**| 40.5|
> |IPMix| **11.1**|**22.8**|10.4|**56.5**|**71.5**|27.2| **41.1**|
>
> From the experimental results, overall, IPMix has achieved better results against different data shifts. However, it only achieved suboptimal performance on ImageNet-style and ImageNet-3DCC. We will explore the reasons and optimize our method in future work.
>
> Q4: Due to the space limitation of the main paper, we have to put part of the content in the appendix. Meanwhile, as NeurIPS requires separate submissions for the main paper and appendix, direct links within the main text were not feasible. We apologize for any inconvenience caused.
>
> Q5: We have added experiments on the transformer architecture, please see results in the "global" response.
>
> Due to the word limit in the rebuttal, we couldn't address all concerns exhaustively. Please let us know if you have any further questions.
>
> [A] Sariyildiz et al. Fake It Till You Make It: Learning Transferable Representations From Synthetic ImageNet Clones, CVPR 2023.

---

> > ### Comment · Reviewer_jACH · 2023-08-17
> > **Thanks**
> >
> > I have read other reviews and the rebuttal. I would like to thank the authors for their effort. The rebuttal was helpful, thus I am increasing my rating to borderline accept. Please include the new results in the updated version of your manuscript (Full ImageNet results could be more useful for the robustness & safety researchers to truly assess the capability of this method).

---

> > > ### Author Response · Authors · 2023-08-17
> > > **Thanks to Reviewer jACH**
> > >
> > >
> > > Thank you for taking the time to re-evaluate our submission and for recognizing our efforts in the rebuttal. We genuinely appreciate your constructive feedback. Please be assured that we will include the new results in our revised manuscript.
> > >
> > > Once again, thank you for your insights and suggestions that have contributed to improving our paper.

---

### Official Review · Reviewer_kzv7 · 2023-07-03

**Soundness:** 3 good
**Presentation:** 3 good
**Contribution:** 2 fair
**Rating:** 4
**Confidence:** 4

**Summary:**

This work proposes a data augmentation method, named IPMix, which is designed to improve robustness without hurting clean accuracy. Specifically, IPMix utilizes different levels of data augmentation (image-level, patch-level, and pixel-level) to increase the diversity of training data and introduces structural complexity to generate more diverse images. The authors have evaluated the robustness of IPMix to adversarial perturbations, calibration, prediction consistency, and anomaly detection.

**Strengths:**

1. The evaluation of robustness and ablation study is extensive. The authors have comprehensively validated the robustness of IPMix, including adversarial perturbations, calibration, prediction consistency, and anomaly detection. Moreover, the ablation of design choices is in detail.
2. The results on small-scale datasets are promising. Overall, IPMix achieves promising accuracy on CIFAR datasets, as well as safety metrics.

**Weaknesses:**

1. Comparisons with RandAugment [1]. The whole design of IPMix seems to be a complicated combination of multiple existing data augmentation methods and the image-level transformations are important components of RandAugment. According to Tab. 6, these image-level transformations are the most effective modules in the whole design and it is suggested to compare IPMix with RandAugment, which is one of the most popular choices in current training frameworks.
2. Generalization ability to competing methods. Current training frameworks of object recognition [2][3] often utilize multiple data augmentation methods during training, the reviewer wonders whether IPMix is compatible with these methods and could further improve the performance.
3. Results on ImageNet. The accuracy of IPMix on imageNet is not very convincing and the whole results on ImageNet seem strange, as Mixup, CutMix both lead to inferior results compared to the baseline. This phenomenon is contradictory to the results of their papers and the explanation of this part is highly appreciated.

[1] Cubuk E D, Zoph B, Shlens J, et al. Randaugment: Practical automated data augmentation with a reduced search space[C]//Proceedings of the IEEE/CVF conference on computer vision and pattern recognition workshops. 2020: 702-703.
[2] Liu Z, Mao H, Wu C Y, et al. A convnet for the 2020s[C]//Proceedings of the IEEE/CVF conference on computer vision and pattern recognition. 2022: 11976-11986.
[3] Li K, Wang Y, Gao P, et al. Uniformer: Unified transformer for efficient spatiotemporal representation learning[J]. arXiv preprint arXiv:2201.04676, 2022.

**Questions:**

Apart from the points in Weaknesses, the reviewer also has some other questions:

1. Validation across different architectures. The authors have only validated IPMix on ResNet which not be enough to prove the generalization ability of IPMix.
2. Revision on Fig. 4. It is advised to add the names of the three methods from Tab. 2 to Fig. 4 which will be easier for reading.

---

> ### Author Rebuttal · Authors · 2023-08-09
>
> Thank you for your valuable time in reviewing our paper.
>
> W1: Thank you for your valuable suggestion.
>
> We agree that RandAugment is a popular image-level method and believe that comparing with it can strengthen our work. While space constraints in our main paper limited a direct comparison with RandAugment, we did feature it as a representative image-level method in Figure 2. Additionally, in Appendix B.2, we contrasted IPMix with other prevalent image-level techniques, such as AutoAugment, RandAugment, and TrivialAugment, underscoring the advantages of our method. For a more comprehensive review, we've summarized the content from Appendix B.2, which evaluates Accuracy, Robustness, Consistency, Calibration, and Adversaries on CIFAR100 with WRN-28x10. Mean and standard derivation over three random seeds is shown for each experiment.
>
> ||Classification (Error)↓|Robustness (mCE)↓ |Adversaries (Error)↓|Consistency (mFR)↓|Calibration (RMS)↓|
> |-|-|-|-|-|-|
> |AutoAugment|17.7 $ \pm $0.11|38.4 $ \pm $0.15|97.8 $ \pm $0.22|8 $ \pm $0.06|7.9 $ \pm $0.06|
> |RandAugment|17.8 $ \pm $0.14|41.5 $ \pm $0.13|96.6 $ \pm $0.25|8.6 $ \pm $0.10|7.9 $ \pm $0.04|
> |TrivialAugment|18$ \pm $0.13|35.4 $ \pm $0.21|96.3 $ \pm $0.23|7.3 $ \pm $0.07|8.7 $ \pm $0.04|
> |IPMix|**17.4 $ \pm $0.17**|**26.6 $ \pm $0.19**|**91.3 $ \pm $0.21**|**4.2 $ \pm $0.11**|**6.4 $ \pm $0.07**|
>
> W2: Thank you for your questions about the generalization ability of IPMix. To address this, we conducted additional experiments on UniFormer and ConvNeXt, as you mentioned. Due to time and resource constraints, we conducted our experiments on smaller versions of the corresponding models. Specially, we trained UniFormer-XXS (224x224) from scratch for 300 epochs and fine-tuned ConvNeXt-Tiny for 90 epochs. Below, we have provided the relevant experimental results. These results demonstrate the generality and effectiveness of IPMix.
>
> ||Classification (Accuracy)↑|ImageNet-C (Error)↓ |ImageNet-R (Error)↓|Calibration (RMS)↓
> |-|-|-|-|-|
> |UniFormer|80.2|52|57.2|6.1
> |UniFormer+IPMix|**80.3 (↑0.1)**|**43.8 (↓8.2)**|**56.5 (↓0.7)**|**5.9 (↓0.2)**
> |ConvNeXt|81.5|43|52.5|7.2
> |ConvNeXt+IPMix|**81.8 (↑0.3)**|**36.2 (↓6.8)**|**50.8 (↓1.7)**|**7.1 (↓0.1)**
>
> W3: Thank you for your feedback.
>
> Firstly, we'd like to point out that the experimental settings in our study differ from the original papers. In the original works, Mixup trained on ResNet-50 and ResNet-101 for 200 epochs, whereas CutMix trained for 300 epochs. In contrast, our experiments were conducted on ResNet-18, with only 90 epochs due to computational constraints and for easier tuning. This variation in settings can lead to different outcomes, as the **data complexity** and **model capacity** play vital roles in determining performance.
>
> Diving deeper from a theoretical perspective, the VC dimension of a model, denoted as $D$, sets the boundary for the maximum dataset size it can "perfectly" classify.  For the generalization error on test data to be close to its training error, we typically require the number of training samples, $n$, to satisfy: $ n = O_{\gamma, \delta}(D) $, where $D$ is the VC dimension of the model, and $γ$ is the desired error boundary. However, data augmentation techniques like Mixup and CutMix increase the effective training sample count, $n$, by diversifying and complicating the data.  It's pivotal to recognize that this doesn't inherently increase the model's capacity in terms of its VC dimension.  *Consequently, while our training size, $n$, has expanded, the model's capability may not match this increase, potentially leading to **underfitting***. This relationship emphasizes that the augmented number of training samples, $n$, must align with the model's VC dimension $D$ to ensure adequate performance.
>
>  Moreover, data augmentation techniques, such as Mixup and CutMix, enhance the complexity of training data, expanding and complicating the data's feature space. However, these augmentation methods may introduce a **"manifold intrusion"** problem during data mixing. *In essence, generating new samples with soft labels may conflict with the original ones across data manifolds*. Please refer to the "global" response for more details.
>
> Therefore, we face a dual challenge: on one hand, the augmentation methods increase data complexity, requiring a model with a higher VC dimension for a perfect fit; on the other hand, due to manifold intrusion, we may now be training on a less authentic data distribution. Thus, when a model's complexity isn't sufficient to capture this enhanced data complexity, such as ResNet-18, it might lead to **underfitting**.
>
> It's important to note that simply increasing the training epochs won't fix this underfitting. For instance, even if MixUp and CutMix are applied for 300 epochs on ResNet-18, their performance might still not surpass Vanilla, as shown in the table follows:
>
> |Backbones|Beta|ResNet18 |
> |-|-|-|
> |Epochs|$\alpha$|300 epochs
> |Vanilla|-|71.83
> |MixUp|0.2|71.72
> |CutMix|1|71.01
>
> **Note: You can find these results from https://github.com/Westlake-AI/openmixup/tree/main/configs/classification/imagenet/mixups/**
>
> Hence, our experimental results are in line with your intuition. When we employ data augmentation techniques, we must consider multiple factors such as the **model's size**, **dataset size**, and **hyperparameter settings**. Methods such as CutMix and MixUp may only demonstrate their efficacy on ResNet-50 or **larger models**.
>
> Q1: We have added experiments on the transformer architecture, please see results in the "global" response.
>
> Q2: We have modified it according to your suggestion, please see the revised Figure 4 in the "global" response PDF.
>
> Due to the word limit in the rebuttal, we couldn't address all concerns exhaustively. Please let us know if you have any further questions.

---

> > ### Comment · Reviewer_kzv7 · 2023-08-15
> > **Reply to rebuttal**
> >
> > Thanks for the detailed reply from the authors and some of the concerns are addressed.
> >
> > However, the provided results on ImageNet are not very promising to the reviewer since the improvement is very limited and nearly all the results on ImageNet are obtained using tiny networks. As the authors explained in W3: the same method may exhibit different performances with different model capacities, the efficacy of IPMix on larger models and large-scale datasets remains a concern.
> >
> > After reading all the replies, the reviewer does feel that the efficacy of IPMix on large-scale datasets like ImageNet is not convincing enough and the technical contribution is also unclear. Therefore, the reviewer decides to keep the original rating.

---

> > > ### Author Response · Authors · 2023-08-15
> > > **Thanks for your reply**
> > >
> > > Thank you for the feedback on our submission.
> > >
> > > 1. We understand your concerns regarding the application of IPMix with larger models, and we'd like to address them as follows:
> > >
> > > * **Performance on Larger Models**: In our supplementary experiments with models like ConvNeXt-T and Swin-Tiny, which are **comparably sized** to the widely used ResNet-50 (ConvNeXt-T: 28M, Swin-Tiny: 28M, ResNet-50: 25.6M), IPMix has demonstrated significant performance gains. This indicates the **generalization ability** of our method and its potential applicability to larger architectures.
> > >
> > > **Experiments on ImageNet:**
> > >
> > > | |Classification (Accuracy)↑|	ImageNet-C (Error)↓|	ImageNet-R (Error)↓|	Calibration (RMS)↓|
> > > |--|-|-|-|-|
> > > ConvNeXt-Tiny (28M)	|81.5 |	43|	52.5|	7.2|
> > > ConvNeXt-Tiny + IPMix  | **81.8 (+0.3)** |	**36.2 (-6.8)** |	**50.8 (-1.7)** |	**7.1 (-0.1)** |
> > > Swin-Tiny (28M)	|80.9	|47.1	|58.3	|7.3|
> > > Swin-Tiny + IPMix  |**81.2(+0.3)** |	**41.3(-5.8)** |	**56.5(-1.7)** | **6.7(-0.6)** |
> > >
> > > * **Experiments on Small Models:** Due to computational and time constraints, we conducted many of our experiments on smaller models. However, it's worth noting that the improvements we achieved, especially on ImageNet, are **not minor**. We observed improvements across various safety metrics without compromising the model's accuracy, which is more meaningful from a practical perspective.
> > >
> > > * **Plug-and-Play Nature on ImageNet:** Our supplemental experiments have shown that IPMix exhibits strong **plug-and-play** when integrated with existing models, significantly enhancing their safety performance. We believe this attribute is **model-size agnostic**. Additionally, it's essential to recognize the efficiency of improving smaller models to match the performance of larger ones. For instance, a ResNet-18 enhanced with IPMix outperforms a vanilla ResNet-50 on ImageNet-C (ResNet-18+IPMix: **77.8** mCE, ResNet-50: 78.2 mCE), showcasing potential **resource savings**.
> > >
> > > * **Future Work on Bigger Architectures:** We are in the process of conducting experiments with more extensive models, including ResNet-50. We plan to integrate these results into the revised manuscript to address concerns about IPMix's effectiveness on larger models.
> > >
> > > 2. We apologize if the technical contribution appeared unclear. Our main contribution are follows:
> > >
> > > * **Improving Diversity and Complexity**: While data diversity and structural complexity are keys to model robustness, most previous methods to improve robustness focus on either generating diverse samples or increasing complexity. We are the **first** to propose a data augmentation method that mixes fractals with the original image at different levels and combines the strengths of these methods into a unified and coherent whole, to enhance the **overall performance**.
> > >
> > >  * **Comprehensive Improving Safety Metrics:** Most methods target only accuracy or data shifts from corruption. In contrast, we aim to elevate performance across **multiple safety metrics**, which is more meaningful from a practical perspective.
> > >
> > > * **A Unique Paradigm:** Although some components of our approach have been proposed before, we believe that the paradigm of our entire method is **unprecedented**. For example, although the paper [1] introduced Random Pixels and Random Elements, the reported experimental results indicated that these methods achieved the lowest accuracy (4.3%↓ and 4.5%↓ by comparing with Mixup). We discover that mixing the original image and fractals using these two methods can **increase the robustness without sacrificing accuracy**. Our motivation is to combine these components that would typically affect the accuracy to showcase potential performance. We posit that this task is **non-trivial**, but rather holds significant value and can benefit future research.
> > >
> > > We will strive to make this clearer in our future revisions.
> > >
> > > We genuinely value your insights and will continue to enhance our work, keeping your feedback in mind.
> > >
> > > [1] Summers, Cecilia, and Michael J. Dinneen. "Improved mixed-example data augmentation." In 2019 IEEE winter conference on applications of computer vision (WACV), pp. 1262-1270. IEEE, 2019.

---

> > > ### Author Response · Authors · 2023-08-16
> > > **An extra supplement**
> > >
> > > We would like to share a critical observation from our experiments. While the combination of MixUp, CutMix, and AutoAugment is widely adopted as a default data augmentation strategy in many models, our findings suggest that merely combining multiple augmentation levels (pixel, patch, and image) might not always be optimal and can potentially degrade a model's safety performance. This observation is corroborated in our experiments with DeiT: when we **replaced** its default data augmentations (MixUp+CutMix+AutoAugment) with IPMix, the model's performance significantly improved. Hence, we believe that delving deeper into this area holds substantial value.
> > >
> > > | |Classification (Accuracy)↑|	ImageNet-C (Error)↓|	ImageNet-R (Error)↓|	Calibration (RMS)↓|
> > > |--|-|-|-|-|
> > > DeiT	|72	|57	|66.8	|13.3|
> > > DeiT + IPMix  | **72.6 (+0.6)**	|**50.4 (-6.6)**	|**64.1 (-2.7)**	|**10.5 (-2.8)**|

---

> > > ### Author Response · Authors · 2023-08-21
> > > **The results of IPMix on ImageNet**
> > >
> > > To better demonstrate the efficacy of IPMix on large-scale datasets such as ImageNet, especially with larger models, we've conducted additional experiments using ResNet-50. We trained the model for 180 epochs, adhering to the standard training scheme outlined in [1]. Due to computational and time limitations, we focused our experiments on IPMix, Vanilla, and AugMix (with results taken from the original paper). We firmly believe that these results provide a more compelling argument for the effectiveness of IPMix with bigger models on ImageNet.  Furthermore, please be assured that the complete results will be included in the revised manuscript.
> > >
> > > **Experiments on ImageNet with ResNet-50:**
> > >
> > > | | Classification (Error)↓ | ImageNet-C (mCE)↓ | ImageNet-P (mFR)↓ |Calibration (RMS)↓ |
> > > |--|-|-|-|-|
> > > | Vanilla          |             23.9 	|80.6	|57.2	|5.6
> > > | AugMix         |            22.4 	|68.4	|37.4	|4.5
> > > | IPMix          |             **22.1** 	|**64.5**	|**35.7**	|**3.2**
> > >
> > > Please note that OpenReview will not allow us to address additional questions or concerns after tomorrow. Are there any further experiments you'd like us to perform, or do you have any other concerns?
> > >
> > > Once again, thank you for your invaluable insights and queries during the rebuttal phase. Your feedback has undeniably helped enhance our work.
> > >
> > > [1] Goyal, Priya et al. “Accurate, Large Minibatch SGD: Training ImageNet in 1 Hour.” ArXiv abs/1706.02677 (2017): n. pag.

---

### Official Review · Reviewer_jXDs · 2023-07-03

**Soundness:** 3 good
**Presentation:** 3 good
**Contribution:** 3 good
**Rating:** 7
**Confidence:** 4

**Summary:**

The authors propose IPMix -- a new data augmentation technique applied during training that improves model robustness across a variety of test-time tasks (e.g., distributional shift, adversarial data augmentation) without compromising accuracy on in distribution tasks (when compared to a model trained in a traditional manner). The acronym IPMix indicates that the technique utilizes augmentations at the Image (I) and Pixel/Patch (P) level and mixes these together with each other and the original image. IPMix augments samples via a convex combination of an augmented and original image, where the novelty in the augmentation operations applied to the image (compared to existing augmentation techniques e.g. Augmix) come from operations between an image and unlabeled synthetic images (e.g., fractals). IPMix research is motivated to produce safe ML systems (e.g., improve model robustness and calibration).

**Main Contributions:**

* IPMix: A label preserving data augmentation technique that achieves most balanced performance across a variety of tasks desired for safe ML systems (accuracy, robustness, calibration, and anomaly detection).
* A thorough empirical study of IPMix using multiple datasets, architectures, and tasks compared to large set of existing data augmentation schemes.

**Post-rebuttal update**
The authors addressed all of my weaknesses and questions and I have raised my score from a weak accept to accept.

**Strengths:**

* **Experiments**: Experimental results with IPMix compared to existing data augmentation methods (in terms of model accuracy, robustness, and calibration) are impressive and are largely consistent across many datasets and architectures.
* **Data augmentation analysis**: Some analysis of models trained with varying data augmentation schemes highlights strengths of IPMix (Figure 8) in comparison to existing data augmentation schemes.

**Weaknesses:**

-	**Motivation**: The use of unlabeled synthetic images in the Pixel/Patch mixing did not seem well motivated to me. I agree that IPMix seems to work well based on the experimental results and understand that the choice of unlabeled synthetic images avoids mislabeled augmented training images. However, I feel that some motivation (practical or theoretical) is needed to help clarify and better motivate the design of IPMix and why it would improve performance beyond existing methods (e.g., Augmix) that do not produce augmented images with incorrect labels.
-	**Paper organization**: A lot of the paper is relegated to the appendix (Appendix F (combination experiments), Appendix E (Fractal generation), Appendix B.1 (IPMix insensitive to mixing set change), Appendix C (IPMix algorithm). I understand space constraints and submission limitations, but I would have liked to see more of the details of IPMix and experiments informing its design in the main body. Perhaps the related work section could have been in the appendix to make space for this.

**Questions:**

1.	**Motivation**: What is the motivation for unlabeled synthetic images (e.g., fractals) used in IPMix set? I suspect this is very clear to the authors, but it is not well motivated or explained in the main body.
2.	**Experiments**: It is unclear from the paper and appendix if multiple replicates are trained for each architecture/dataset/augmentation configuration (Tables 3, 4, and Figure 6). I am under the impression that a single model was trained for each configuration. If that is the case, it is unclear how much deviation (in accuracy, robustness, calibration) the models trained with varying data augmentation schemes have. I would typically expect some result on this for smaller datasets (e.g., CIFAR-10). Is this something you considered that I missed?
3.	**Overhead/cost**: Is there an observable overhead/cost of using IPMix during training (when compared to vanilla training or other data augmentation techniques)? In my experience, Augmix increases training time due to augmentation scheme so I am curious if IPMix is more/less costly than Augmix (as it produces better models than Augmix).
4.	**Robustness**: How do models trained using IPMix compare to robustbench leaderboard Common corruptions (i.e., CIFAR-10-C, ImageNet-C)? I suspect they are comparable, but it would potentially strengthen the work by comparing to SOTA models from robustbench. The architectures used in the paper are commonly used on robustbench so it would likely be an easy/straightforward comparison. *In this direction: Did you consider utilizing the Jensen-Shannon divergence (JSD) loss utilized in the original Augmix paper in experiments with IPMix? I suspect this may offer additional improvements to models trained with IPMix so it may be worthwhile to consider if comparing to robustbench models.*
5. **Figure 6**: Is this calibration error for a single architecture trained using each data augmentation technique evaluated on CIFAR-100 (e.g. ResNet-18)? The caption does not specify.

Minor suggestions:
* Table 1 is provided but never referenced or commented on in the text of the paper. I would typically expect some minor commentary on tables/figures.
* Table 1: “Accuracy Error” seems like odd/confusing phrasing to me. Wouldn’t “Classification Error” be clearer?

**Limitations:**

Yes

---

> ### Author Rebuttal · Authors · 2023-08-08
>
> Thank you for your valuable time in reviewing our paper.
>
> W1-1: "The motivation of using synthetic images. "
>
> Due to the difficulty in acquiring labeled data, as well as human bias and data privacy issues, more and more research focuses on obtaining synthetic data that can be generated without labeling. Moreover, most studies concentrate on pre-training on synthetic data and then transferring to downstream tasks to alleviate the model's dependence on labeled data. Among these studies, [3] pointed out that **diversity** and **structural complexity** of synthetic data are keys to excellent performance. Meanwhile, as previously indicated in research [4], these two properties are also vital to enhancing model robustness. Since synthetic data are easy to obtain and mixing with them do not encounter "manifold intrusion", which leads us to question whether synthetic data can be applied to data augmentation at different scales, improving the model's robustness by increasing the image's structural complexity and data diversity. Furthermore, our thoughts are inspired by and tally with **human visual principles**. We continually develop our ability to distinguish various subjects by being exposed to a wide array of objects (diversity) and encountering intricately structured targets (structural complexity). Finally, the significant results of IPMix validate our ideas.
>
> Based on this, we believe the motivation for utilizing unlabeled synthetic data is sufficient and necessary.
>
> W1-2: "Comparison with other label-preserving methods"
>
> Below we will address your concerns by comparing IPMix with AugMix from different views to show why our approach improves performance.
>
> *  **Motivation**: AugMix mixes an image with its versions that have been processed with image-level methods (e.g., rotation) to improve robustness. However, AugMix has lower accuracy compared to other patch-level methods, as shown in Table 1. Our motivation is to combine the strengths of different levels of methods into a unified and coherent whole, to enhance the overall performance.
>
> *  **Objective**: The purpose of AugMix is to tackle data shifts caused by corruption, but the actual situations encountered in real-world scenarios are much more complex, as mentioned in the paper [6]. Our objective is to improve performance across multiple safety metrics simultaneously, which is more meaningful from a **practical perspective**.
>
> *  **Method**: Except for the image-level augmentation, IPMix mixes clean data with fractals at different levels to improve **structural complexity** and uses **random pixels mixing** and **random elements mixing** for better data integration. While AugMix applies augmentation transformations only at the image level, its performance in classifying partially occluded objects is sub-optimal.
>
> We firmly believe that the above reasons are the reasons why IPMix performs well.
>
> W2: We genuinely appreciate your suggestion.
>
> We value the thorough acknowledgment of related work as it respects prior research. Meanwhile, we recognize and agree with your suggestion to detail more of IPMix in the main paper. We will strive to incorporate more details while adhering to page limits in the revised manuscript.
>
> Q1: Please refer to W1 for detailed response.
>
> Q2: We apologize for the oversight. While multiple replicates were conducted, only the ablation study in Table 6 explicitly mentions this. Due to space constraints, we only present the repeated results of PixMix, AugMix, and IPMix on CIFAR-10 here, and we will ensure that the revised manuscript includes all the results.
>
> **Experiments on CIFAR-10 with WRN28-10:**
>
> ||Classification (Error)↓|Robustness (mCE)↓ |Adversaries (Error)↓|Consistency (mFR)↓|Calibration (RMS)↓|
> |-|-|-|-|-|-|
> |AugMix|3.4 $ \pm $0.16|9.1 $ \pm $0.21|81.9 $ \pm $0.21|1.2 $ \pm $0.11|3.8 $ \pm $0.13|
> |PixMix|3.8 $ \pm $0.20|8.7 $ \pm $0.21|92.9 $ \pm $0.25|1.6 $ \pm $0.09|2.9 $ \pm $0.21|
> |IPMix|**3.3 $ \pm $0.17**|**7.5 $ \pm $0.23**|**76.4 $ \pm $0.23**|**1.1 $ \pm $0.10**|**1.9 $ \pm $0.16**|
>
> Q3: Thank you for your question. We have addressed the overhead of using IPMix during training in Appendix G. Please refer to it for a detailed response.
>
> Q4-1: It's worth noting that direct comparisons with other methods might be subject to discrepancies due to factors such as epochs, architecture, and the use of extra data. Based on robustbench, for CIFAR-10-C, the Winning Hand achieved a Robustness accuracy of **92.78%** on WRN28-4. In contrast, IPMix achieved **91.4%** and **92.5%** on WRN40-4 and WRN28-10, respectively, and please note that we only trained for 100 epochs. For CIFAR-100 -C, while Winning Hand reached **71.08%** accuracy on WRN28-4, our method surpassed this with **71.4%** and **73.4%** on WRN40-4 and WRN28-10, respectively, which significantly outperforms the current best approach. As for ImageNet-C, given that our ResNet18 model is considerably smaller than ResNet50, we haven't made a comparison. We aim to explore this direction in future experiments and believe that the existing results are sufficient to prove the significance of IPMix. Please note that our goal is to improve overall safety metrics.
>
> Q4-2: "JSD loss"
>
> We did experiments with the JSD-loss and found it can improve performance. However, using JSD loss requires at least three times the memory per batch, which can significantly extend the training time. Hence, we consider treating JSD as a hyperparameter and add analysis and experiments to the revised manuscript.
>
> Q5: The calibration error was evaluated on WRN40-4. Please refer to the "global" response PDF for the revised Figure 6.
>
> M1: We will add data analysis in the revised manuscript. Thanks.
>
> M2: Your suggestion is incorporated in the new version of the paper. Thank you.
>
> Due to the word limit in the rebuttal, we couldn't address all concerns exhaustively. Please let us know if you have any further questions.
>
> Please refer to relevant references in the "global" response.

---

> > ### Comment · Reviewer_jXDs · 2023-08-19
> >
> > Thank you for your thorough response. I have read your individual and global response and appreciated the detail, sourcing, and thoroughness. You have addressed my weaknesses and questions. Accordingly, I am raising my score from weak accept to accept with the expectation that the authors will make the revisions they indicated in my individual response.

---

> > > ### Author Response · Authors · 2023-08-19
> > > **Thanks to Reviewer jXDs**
> > >
> > > Thank you for reconsidering our paper and acknowledging our work during the rebuttal. We're glad to hear that our responses have addressed your concerns. Please rest assured that we will incorporate the revisions as indicated in our response to your comments.
> > >
> > > We greatly appreciate your feedback and support in improving our paper!

---

### Official Review · Reviewer_Bz96 · 2023-07-06

**Soundness:** 2 fair
**Presentation:** 2 fair
**Contribution:** 2 fair
**Rating:** 6
**Confidence:** 3

**Summary:**

This paper proposes a new data augmentation method, IPMix, for vision models that integrates three levels of augmentation techniques (image level, patch level, and pixel level). The approach is label preserving and enforces extra structural complexity to enhance the robustness and other safety measures. Experimental results on multiple open datasets and backbone models are provided to support the research claims in the paper.

**Strengths:**

1. In addition to clean accuracy, the authors seek to optimize both the model robustness and various safety measures in the paper.

2. The provided evaluations are extensive and improvements are observed in multiple performance perspectives compared to the baselines.

3. The paper is generally easy to understand and follow.

**Weaknesses:**

1. Throughout the whole paper, it is kind of hard to understand what is the main technical contribution made in the paper that is original and novel. Although multiple components have been introduced, they seem to be trivial applications and combinations of previously published approaches, since the design of each component of inspired by previous works. Besides, those components are not connected and introduced through a coherent idea but are more like random pieces that are very loosely connected. What main technical challenges have been solved in this paper? If we consider the combination pipeline of different augmentations as the main contribution, it is still trivial because the authors just experimented with different intuitive combination methods in Table 2 and selected the overall best one.

2. From the paper, I don't understand exactly why label-variant augmentations lead to manifold intrusion and lower robustness against unseen data. If the label-preserving property is important, can we enhance the CutMix and MixUp by mixing original images with fractal images and achieving a similar performance as IPMIx?

3. In the evaluation section, the authors only focus on showing the results but do not provide any analysis of the results, which results in the fact that the only piece of information expressed there is IPMix is better than the baselines in various settings, but not about why IPMix works better.

4. The ablation study is not designed in the correct way. Its current form only conveys the message that all three levels of augmentations are needed while removing any one of them will lead to degradations in the performance. In addition to that, the authors should use experiments to show the performance degradation without proper combinations, fractals (label variance), and the pieces in Section 4.2.

5. The authors mentioned the computation efficiency improvement caused by the approach, but no evaluations are performed to test the computation efficiency between IPMix and conventional image-level augmentations.

6. Here are some minor comments:

    (1) Since the paper depends on image-specific intuitions, it would be good to be precise on the scope in the paper title.

    (2) In line 183, the \lambda is directly used without introducing its purpose and functionality.

    (3) The specific design scope and definition of IPMix is unclear. The title of Section 4.1 is about IPMix, while Section 4.2 further introduces more design components. Do they belong to part of IPMix? If so, the organization of the paper should be fixed.

    (4) The ticks in Figure 6 are too small and hard to recognize.

**Questions:**

I think the main question that should be answered is the main technical contribution made in this paper compared to the existing literature, and what main challenges have been solved.

**Limitations:**

Yes, the limitations are discussed in the Appendix of the paper.

---

> ### Author Rebuttal · Authors · 2023-08-08
>
> Thank you for your valuable time and detailed reviews.
>
>  Below we will address your concerns about novelty and what problem we solved.
>
> W1-1: "novelty and technical contributions."
> * **Diversity & Complexity**: While data diversity and structural complexity are keys to model robustness, most previous methods to improve robustness focus on either generating diverse samples or increasing complexity. We are the **first** to propose a data augmentation method that mixes fractals with the original image at different levels and combines the strengths of these two methods into a unified and coherent whole, to enhance the overall performance.
> * **Comprehensive Safety Metrics**: Most methods target only accuracy or data shifts from corruption. In contrast, we aim to elevate performance across multiple safety metrics, which is more meaningful from a practical perspective.
> *  **Unique Paradigm**: Although some components of our approach have been proposed before, we believe that the paradigm of our entire method is **unprecedented**. For instance, while paper [5] introduced Random Pixels and Random Elements with notably lower accuracy, we found that mixing original images with fractals via these methods enhances robustness without compromising accuracy. Our motivation is to combine these components to showcase potential performance. We posit that this task is non-trivial, but rather holds significant value and can benefit future research.
>
> W1-2: 'what main challenges have been solved."
>
> **Diverse Distribution Shifts:** We improved the model's ability to resist distribution shifts at various levels.
>
> **Holistic Safety Performance:** Our approach offers a comprehensive improvement across multiple safety metrics without sacrificing accuracy.
>
> **Research Direction:** We pioneered a direction for exploration wherein by mixing original images with synthetic images in a label-preserving way, we've amplified data's structural complexity, thus elevating overall model performance.
>
> Based on the aforementioned statements, we firmly believe that IPMix is both novel and significant.
>
> W2-1: "The drawbacks of label-variant methods. "
>
> Please see the detailed answer in the "global" response.
>
> W2-2: "What if mixing MixUp and CutMix with fractals in a label-preserving way?"
>
> Please note that MixUp regularizes the neural network to favor simple linear behavior in-between training examples, and CutMix aims to recognize two training objects from their respective partial views. If we directly mix them with fractals outside the dataset, it may violate the original intention of these methods. Nevertheless, we still considered two experimental scenarios: 1. Mixing the original image with fractals first, then leveraging MixUp/CutMix based on the generated image in a **label-variant** way. 2. Directly mix the original image with fractals in a **label-preserving** way. Due to limited space, we only show the second solution the result of.
>
> **Experiment on CIFAR-100 with WRN40-4:**
>
> ||Classification|Robustness|Adversaries|Consistency|Calibration|
> |-|-|-|-|-|-|
> |MixUp|20.5 $\pm $0.11|45.9 $\pm $0.12|96.9 $\pm $0.14|8.9 $\pm $0.04|10.5 $\pm $0.02
> |MixUp + fractal|20.6 $\pm $0.19|46.5 $\pm $0.23|97.1 $\pm $0.21|9.9 $\pm $0.09|11.3 $\pm $0.04
> |CutMix|20.3 $\pm $0.08|50 $\pm $0.09|96.9 $\pm $0.12|12 $\pm $0.05|10.5 $\pm $0.05
> |CutMix + fractal|20.8 $\pm $0.13|50.3 $\pm $0.21|97.0 $\pm $0.16|10.7 $\pm $0.11|14.4 $\pm $0.08
> |IPMix|**19.4 $\pm $0.17**|**28.6 $\pm $0.2**|**89.4 $\pm $0.18**|**4.3 $\pm $0.09**|**2.8$\pm $0.07**
>
> The experiment shows that directly mixing the original image with fractals in a label-preserving way will degrade model performance. This may stem from the added structural complexity failing to offset the loss of inductive bias.
>
> W3: We provide an analysis of why IPMix outperforms the other augmentation methods in Section 6. We apologize for the misleading in the evaluation section and will modify relevant content in the revised manuscript.
>
> W4:  Thank you for the valuable feedback.
>
> We would like to kindly point out that these concerns have been addressed in Appendix B. Due to space constraints, we placed the ablation experiments in the Appendix and apologize for any misleading.
>
> W5:
> IPMix offers significant computational advantages by requiring only training once. In contrast, conventional image-level methods demand an extensive search for the optimal augmentation policy, as noted in [1,2]. The table below further illustrates the search overhead difference between IPMix and image-level methods. The data are from the original paper and vary depending on the dataset used (CIFAR, SVHN, and ImageNet).
>
> |Method| Serach Overhead |
> |-|-|
> |AutoAugment|40-800x
> |RandAugment|4-80x
> |Fast AutoAugment|1x
> |IPMix|**0x**
>
> W6(1): Your suggestion is incorporated in the new version of the paper. Thank you.
>
> W6(2):  Thank you for pointing that out. In the equation provided, \lambda represents the mixing ratio, derived from the Beta distribution ($\alpha=1$ by default), indicating the proportion of the content taken from the input image x1 and the synthetic image x2 when performing mixing operations. We ensure clear representation in the revised manuscript.
>
> W6(3): In our manuscript, Section 4.1 presents the core framework of IPMix, while Section 4.2 introduces two methods designed for more effective information fusion. These two parts contribute to the excellent performance of IPMix. To provide better clarity, we plan to adjust the section titles as follows:
>
> Section 4: IPMix
>
> Section 4.1: Integrating Different Levels into a Coherent Approach
>
> Section 4.2: Multi-Scale Information Fusion
>
> We appreciate your feedback once again.
>
> W6(4): Please refer to the "global" response PDF for the revised Figure 6.
>
> Due to the word limit in the rebuttal, we couldn't address all concerns exhaustively. Please let us know if you have any further questions.
>
> Please refer to relevant references in the "global" response.

---

> > ### Comment · Reviewer_Bz96 · 2023-08-14
> > **Thanks for the thorough respones.**
> >
> > Thanks for the thorough responses. I have read all the replies, and unfortunately, I can not find enough reasons to accept the paper, and can only raise my rating to 4.

---

> > > ### Author Response · Authors · 2023-08-15
> > > **The thanks and further clarification on reviewer Bz96 concerns**
> > >
> > > Thank you for reviewing our paper and taking the time to read our responses. We respect your feedback.
> > >
> > >  We noticed that the main questions: "The main question that should be answered is the main technical contribution made in this paper compared to the existing literature" might not have been addressed in the depth it deserved in our initial rebuttal due to space constraints. To clarify further on this point, we want to compare IPMix with AugMix and other P-level (pixel-level and patch-level) methods.
> > >
> > > 1. Compared with AugMix.
> > >
> > > * **Motivation**: AugMix mixes an image with its versions that have been processed with image-wise (e.g., rotation) methods to improve robustness. However, AugMix has lower accuracy compared to other patch-wise data augmentation methods, as shown in Table 3. Our motivation is to combine the strengths of different levels into a unified and coherent whole, to enhance the **overall** performance.
> > >
> > > * **Objective**: The purpose of AugMix is to tackle data shifts caused by corruption, but the actual situations encountered in real-world scenarios are much more complex than those it takes into account, as mentioned in the paper [1]. Our objective is to improve performance across multiple safety metrics, including corruption robustness, consistency, anomaly detection, adversarial robustness, and calibration confidences simultaneously, which is more meaningful from a **practical perspective.**
> > >
> > > * **Method & Framework**: Except for the image-level augmentation, IPMix mixes clean images with synthetic pictures at different scales to improve **structural complexity** and uses random pixels mixing and random elements mixing for better data integration. While AugMix applies augmentation transformations only at the image level, its performance in classifying partially occluded objects is sub-optimal. Furthermore, our goal is to identify the **optimal framework** for generating highly diverse augmented images. The experimental results in Table 2 imply the potential existence of a **general framework** for data augmentation, which we consider a promising avenue for further research in this field.
> > >
> > > * **Performance**: AugMix is considered a method with state-of-the-art performance in improving model robustness. In comparison with AugMix, our method shows **significant** improvements in CIFAR-10-C, CIFAR-100-C, ImageNet-C, and the other datasets, as shown in Tables 3, 4, and 5 in the main text, which provides compelling evidence of the superiority of IPMix over AugMix.
> > >
> > > 2. Compared with other P-level methods.
> > >
> > > * **Method**: We are the first to propose a data augmentation method that mixes fractals with the original image at different level. Compared with existing P-levels works (CutMix, TokenMix, AutoMix, AdaMixUp,etc), our method is **label-preserving**, which is plug-and-play and significantly improves performance in practice.
> > >
> > > * **Objective**: Our objective for IPMix is to improve model safety and adaptability to real-world complexity. While other P-level methods primarily focus on improving accuracy.
> > >
> > > * **Performance**: Compared to recent patch-level methods that utilize saliency information (e.g., PuzzleMix) to mix samples and labels, our approach achieves better performance without incurring additional computational complexity.
> > >
> > > We list the experimental results of IPMix and other existing methods on CIFAR-100 with WRN28-10 below.
> > >
> > > | | Classification (Error)↓ | Robustness (mCE)↓ | Adversaries (Error)↓ | Consistency (mFR)↓ | Calibration (RMS)↓ |
> > > |--|-|-|-|-|--|
> > > | AutoAugment            |             17.7 $ \pm $ 0.11	|38.4 $ \pm $ 0.15	|97.8 $ \pm $ 0.22	|8 $ \pm $ 0.06	|7.9 $ \pm $ 0.06 |
> > > | RandAugment           | 17.8 $ \pm $ 0.14|	41.5 $ \pm $ 0.13|	96.6 $ \pm $ 0.25|	8.6 $ \pm $ 0.10|	7.9 $ \pm $ 0.04|
> > > | TrivialAugment     |             18 $ \pm $ 0.13|	35.4 $ \pm $ 0.21|	96.3 $ \pm $ 0.23|	7.3 $ \pm $ 0.07|	8.7 $ \pm $ 0.04 |
> > > | SaliencyMix         |            19 $ \pm $ 0.14|	38.3 $ \pm $ 0.24|	96.7 $ \pm $ 0.21|	 10.8 $ \pm $ 0.07|	7.1 $ \pm $ 0.07 |
> > > | PuzzleMix          |             18.3 $ \pm $ 0.11|	 37.9 $ \pm $ 0.21|	96.1 $ \pm $ 0.23|	 10.5 $ \pm $ 0.04|	7.5 $ \pm $ 0.08 |
> > > | Co-Mixup         |   18.1 $ \pm $ 0.19|	35.6 $ \pm $ 0.25|	95.6 $ \pm $ 0.21|	10.1 $ \pm $ 0.05|	7.7 $ \pm $ 0.04 |
> > > | Manifold Mixup         | 18.6 $ \pm $ 0.21|	 51.3 $ \pm $ 0.23|	93.4 $ \pm $ 0.17|	29.9 $ \pm $ 0.28|	10.2 $ \pm $ 0.09 |
> > > | IPMix         | **17.4 $ \pm $ 0.17**|	**26.6 $ \pm $ 0.19**|	**91.3 $ \pm $ 0.21**|	**4.2 $ \pm $ 0.11**|	**6.4 $ \pm $ 0.07**|
> > >
> > > We hope this additional clarification sheds light on the issue. Regardless of the final decision, we genuinely appreciate your insights, which will help improve our work.
> > >
> > > [1] Hendrycks, Dan et al. “Unsolved Problems in ML Safety.” ArXiv abs/2109.13916 (2021): n. pag.

---

> > > ### Author Response · Authors · 2023-08-16
> > > **An extra supplement**
> > >
> > > We would like to share a critical observation from our experiments. While the combination of MixUp, CutMix, and AutoAugment is widely adopted as a default data augmentation strategy in many models, our findings suggest that merely combining multiple augmentation levels (pixel, patch, and image) might not always be optimal and can potentially degrade a model's safety performance. This observation is corroborated in our experiments with DeiT: when we **replaced** its default data augmentations (MixUp+CutMix+AutoAugment) with IPMix, the model's performance significantly improved. Hence, we believe that delving deeper into this area holds substantial value.
> > >
> > > | |Classification (Accuracy)↑|	ImageNet-C (Error)↓|	ImageNet-R (Error)↓|	Calibration (RMS)↓|
> > > |--|-|-|-|-|
> > > DeiT	|72	|57	|66.8	|13.3|
> > > DeiT + IPMix  | **72.6 (+0.6)**	|**50.4 (-6.6)**	|**64.1 (-2.7)**	|**10.5 (-2.8)**|

---

> > > > ### Comment · Reviewer_Bz96 · 2023-08-19
> > > > **Raised my score.**
> > > >
> > > > Thanks for the hard work authors have put into the rebuttal. More of my concerns have been addressed. I am fine with accepting it now. I suggest the authors revise the abstract and introduction in the camera-ready to better reflect the novelty and contributions of the proposal, compared to the existing literature, and add the new results/discussions as they promised.

---

> > > > > ### Author Response · Authors · 2023-08-19
> > > > > **Thanks to Reviewer Bz96**
> > > > >
> > > > > Thank you for recognizing the effort we've put into our work and the rebuttal. We are truly grateful for your constructive feedback and guidance throughout this process.
> > > > >
> > > > > We will carefully consider your suggestions, revising the abstract and introduction to more distinctly emphasize the novelty and contributions of our proposal in the context of existing research. Moreover, we are committed to including the new results and discussions in the final manuscript as promised.
> > > > >
> > > > > Thank you once again for your invaluable insights that have significantly contributed to enhancing the quality of our paper.

---

### Official Review · Reviewer_FKxz · 2023-07-17

**Soundness:** 2 fair
**Presentation:** 3 good
**Contribution:** 2 fair
**Rating:** 6
**Confidence:** 4

**Summary:**

This paper presents IPMix, a new data augmentation strategy that enhances robustness without compromising clean accuracy. The proposed approach diversifies the training data with limited computational overhead. IPMix introduces structural complexity at different levels to generate a wider variety of images and employs a random mixing method for multi-scale information fusion to bolster robustness.

**Strengths:**

- The paper addresses augmentation from both a clean accuracy and safety perspective, which is commendable.
- The paper is generally well-written and easy to follow.
- The figures provided in the paper are informative and well-polished and enhance the understanding of the proposed strategy.
- The paper includes many implementation details and ablation studies in Appendix.

**Weaknesses:**

- The paper could benefit from including error bars to account for variance across different seeds, particularly when the performance of each method is comparable.
- The paper could provide a more in-depth analysis of why the proposed method improves each safety metric. While the explanations regarding diversity and regularization effects are reasonable, it would be beneficial to understand which part of the pipeline is most crucial for a specific safety measure (e.g., robustness, calibration). The significance of increasing structural complexity using "fractals" could be discussed more.
- The title emphasizes the "label-preserving" property, yet it is unclear how vital this is. The authors could discuss the potential outcomes of mixing two images with different labels and linearly interpolating the label.
- The experimental focus is primarily on ResNet variants. Including results on the performance of other architectures (e.g., Transformer) could strengthen the paper.

**Questions:**

- There seems to be no discussion on Table 1, which feels disconnected from the rest of the paper. Could the authors elaborate on this?
- In Section 4.1 "The IPMix framework", the terms "Chain-Mixed", "linear mix", and "mixed input" are not clearly defined. Could the authors clarify these and possibly include the corresponding framework name in Figure 4 alongside the number?
- In Section 3, integration experiments are described. It is suggested that combining previous approaches may result in diminished performance, as the optimal parameter for using each data augmentation independently may not be optimal when used in combination. Could decreasing the augmentation strength of each method yield better performance?
- Table 6 suggests that image-level data augmentation is the most crucial part of CIFAR100. Is this observation consistent with ImageNet as well?
- How can the advantages of the three methods be leveraged while avoiding their drawbacks? Could the authors provide more specific information about the drawbacks of each method?


**Limitations:**

The authors discuss the limitation and societal impact in Appendix.

---

> ### Author Rebuttal · Authors · 2023-08-07
>
> Thanks for your detailed reviews and insightful comments!
>
> W1: Your suggestion is incorporated in the new version of the paper. Thank you.
>
> W2-1: "which component is vital for a specific safety measure."
>
> Thank you for your concerns. We will detailed analyze the impact of each part on different safety metrics through ablation experiment on CIFAR-100.
> |     | Classification | Robustness | Adversaries | Consistency | Calibration |
> |---|------|----|----|--|--|
> | IPMix | **19.4 $\pm $0.17**| **28.6 $\pm $0.2**  | 89.4 $\pm $0.18   | **4.3 $\pm $0.09**                | **2.8 $\pm $0.07** |
> | w/o Patch| 19.7 $ \pm $0.13  | 30 $ \pm $0.21| 91.7 $ \pm $0.15 | 4.7   $ \pm $0.02             | 4.6 $ \pm $0.07|
> | w/o Pixel|19.6 $ \pm $0.09| 33 $ \pm $0.35 | 92.6 $ \pm $0.20 | 5.2 $ \pm $0.05               | 8.2 $ \pm $0.12|
> | w/o Image |20.1 $ \pm $0.27    | 34 $ \pm $0.65 | **87.8 $ \pm $0.22**  | 5.5 $ \pm $0.11               | 8.6 $ \pm $0.21|
>
> **Accuracy**: The **image-level** augmentation has the most substantial effect on accuracy, aligning with current findings[1][2] that image-level methods are commonly used to boost accuracy.
>
> **Robustness**: Both **pixel-level** and **image-level** augmentations improve robustness. Since pixel-level introduces fine-grained variations for pattern recognition, while image-level increases dataset diversity, preventing the model from merely memorizing fixed augmentations.
>
> **Calibration & Consistency**: The **Image-level** part significantly influences calibration and consistency, which increases diversity to improve the prediction calibration across scenarios and ensures consistency in responses to minor perturbations.
>
> **Adversarial Attacks**: Without the **image-level** component, adversarial performance improves, implying diverse data might weaken defense against attacks. Conversely, removing **pixel-level** methods will degrade adversarial robustness, given their inherent resistance to perturbations.
>
> W2-2: The significance of using "fractals."
>
> In recent visual research, given the challenges and cost of acquiring labeled data, there's a growing emphasis on synthetic data generation as an alternative to real data. In this field, the majority of research focuses on pre-training with synthetic data and then fine-tuning for downstream tasks to reduce reliance on costly labeled data. Notably, [3] emphasize that both diversity and **structural complexity** are pivotal for better performance. Moreover, as previously indicated in research [4], these two properties are also vital to enhancing model robustness. Building on this, we want to explore if synthetic data of high structural complexity can be employed for **data augmentation** at different levels to improve model robustness. Significant experimental results with IPMix validate our ideas.
>
> Based on this, we believe that what is truly important is using **synthetic data** to enhance the **structural complexity** of the data. We selected fractals for their unique properties: natural complexity and simple mathematical foundations for image generation. However, our experiments confirm that the performance of IPMix remains relatively stable under the change of synthetic data. Please refer to Appendix B.1 for more details.
>
> W3: "The importance of label-preserving"
>
> Mixing images with distinct labels and linearly interpolating between them will impose certain “local linearity” constraints on the model’s input space beyond the data manifold, which may lead to **"manifold intrusion"**. Consider one experiment on MNIST. If we use MixUp to linearly mix two numbers, such as "1" and "5", the generated image will show the characteristics of "8". When the generated "8" **collides with** a real "8" in the data manifold, there will be a problem of manifold intrusion. Since the two samples have similar characteristics, one is the real label and the other is a soft label ("1" and "5"). This will interfere with its ability to understand and classify categories and degrade model performance.
>
> Building on these finds, we introduced IPMix, which mixes the original image with **unlabeled** synthetic data. This approach increases data diversity and image structural complexity without altering the label, thus enhancing robustness against various data shifts. We believe that this **label-preserving** characteristic is vital to IPMix's outstanding performance.
>
> W4: We have added experiments on the transformer architecture, please see results in the "global" response.
>
> Q1: Thanks for your questions. We discussed Table 1 between L162-L170, where we addressed the phenomenon observed and its potential reasons. The table illustrates that mixing different levels (image, patch, and pixel) sometimes yields worse performance than standalone methods. This could be attributed to the added complexity in the training data, possibly affecting the models' ability to extract meaningful features. For a deeper dive, please refer to Appendix F, which elaborates on this with experimental design.
>
> Q2: The terms are defined in Figure 4's caption. We've updated the figure based on your feedback. Thank you.
>
> Q3:  We have designed experiments based on your inspiration, please view results and analysis in the "global" response PDF, Table 1.
>
> Q4:  Thank you for your question. Regarding ImageNet, we haven't conducted an ablation study due to time and computational constraints. However, based on our CIFAR experiments, we believe image-level augmentation remains crucial for ImageNet. We will add ImageNet results in the new version.
>
> Q5: Please see the detailed answer in the "global" response.
>
> Due to the word limit in the rebuttal, we couldn't address all concerns exhaustively. Please let us know if you have any further questions.
>
> Please refer to relevant references in the "global" response.

---

> > ### Comment · Reviewer_FKxz · 2023-08-14
> > **Thanks for the Rebuttal**
> >
> > I appreciate the authors' efforts in rebuttal. I have read the response carefully. Most of my concerns are addressed, and I hope the authors can incorporate everything they promise into the final version, including the ImageNet study. I am raising my score from 5 (Borderline accept) to 6 (Weak accept). Good luck with your submission.

---

> > > ### Author Response · Authors · 2023-08-15
> > > **Thanks to Reviewer FKxz**
> > >
> > > Thank you very much for taking the time to revisit our submission and for your constructive feedback. We genuinely appreciate your acknowledgment of our efforts in the rebuttal. Please be assured that we are committed to incorporating all the promised improvements into the final version.
> > >
> > >  Thanks again for your insightful responsiveness and for helping us improve our paper!

---

### Author Rebuttal · Authors · 2023-08-09

First, we thank all reviewers for your help and support. Due to the limitation of input characters on OpenReview, we have to put some questions that the reviewers are generally concerned about here and answer them together.

**1. Experiments on Transformer:**

Given the limited time and computational resources available during the rebuttal phase, we strategically conducted three experiments:
* We trained a small ViT for 300 epochs on CIFAR-100 and CIFAR-10. This step aimed to confirm IPMix's potential on smaller datasets using Transformer architectures.
* We initialized DeiT-tiny with IPMix and trained it from scratch for 300 epochs.
* We fine-tuned Swin-Tiny using IPMix for an additional 90 epochs.

These experiments were crafted to showcase the versatility of IPMix across diverse scenarios while highlighting its consistent performance and adaptability within various Transformer architectures.

**Experiments on CIFAR-10:**

||Classification|Robustness|Adversaries|Consistency|Calibration|
|-|-|-|-|-|-|
|Vanilla|19.5 $\pm $0.07|27.7 $\pm $0.14|91.3 $\pm $0.13|5.9 $\pm $0.02|10
|MixUp|19 $\pm $0.11|34.7 $\pm $0.21|89.3 $\pm $0.21|6 $\pm $0.05|9.9 $\pm $0.03
|CutMix|19.3 $\pm $0.08|34.3 $\pm $0.19|89.1 $\pm $0.14|5.5 $\pm $0.05|7.5 $\pm $0.02
|PixMix|28.4 $\pm $0.14|33.9 $\pm $0.24|91 $\pm $0.12|6.5 $\pm $0.11|4.4 $\pm $0.07
|AugMix|20.3 $\pm $0.14|25.6 $\pm $0.2|80.3 $\pm $0.16|5.1 $\pm $0.09|6$\pm $0.08
|IPMix|**19.2 $\pm $0.12**|**23.7 $\pm $0.2**|**75.8 $\pm $0.13**|**3.7 $\pm $0.07**|**5.3$\pm $0.07**

**Experiments on CIFAR-100:**

||Classification|Robustness|Adversaries|Consistency|Calibration|
|-|-|-|-|-|-|
|Vanilla|40.1 $\pm $0.12|56.3 $\pm $0.1|96.2 $\pm $0.14|12.4$\pm $0.04|14.8$\pm $0.02
|MixUp|40 $\pm $0.14|56 $\pm $0.18|92.5 $\pm $0.18|9.8$\pm $0.03|9.5 $\pm$0.02
|CutMix|39.5 $\pm $0.11|56.3 $\pm $0.15|96.2 $\pm $0.17|10$\pm $0.03|9.8 $\pm$0.03
|PixMix|48.7 $\pm $0.14|54.3 $\pm $0.21|93.2 $\pm $0.14|10.9$\pm $0.07|4.9 $\pm$0.04
|AugMix|35.3 $\pm $0.17|42.4 $\pm $0.21|84.6 $\pm $0.16|6.9$\pm $0.03|6.4$\pm$0.07
|IPMix|**32.6 $\pm $0.11**|**39.6 $\pm $0.23**|**83.2 $\pm $0.15**|**6.3$\pm $0.04**|**5.3$\pm $0.05**

**Experiments on DeiT-tiny:**

|Method|Classification (Accuracy)↑|ImageNet-C (Error)↓|ImageNet-R (Error)↓|Calibration (RMS)↓
|-|-|-|-|-|
|DeiT-T|72|57|66.8|13.3|
|+IPMix|**72.6(+0.6)**|**50.4(-6.6)**|**64.1(-2.7)**|**10.5(-2.8)**|

**Experiments on Swin-tiny:**

|Method|Classification (Accuracy)↑|ImageNet-C (Error)↓|ImageNet-R (Error)↓|Calibration (RMS)↓
|-|-|-|-|-|
|Swin-T|80.9|47.1|58.3|7.3|
|+IPMix|**81.2(+0.3)**|**41.3(-5.8)**|**56.5(-1.7)**|**6.7(-0.6)**|

**2. The drawbacks of three methods:**

**The drawbacks of label variant methods:**

* **Pixel-level:** Mixing images with distinct labels and linearly interpolating between them will impose certain “local linearity” constraints on the model’s input space beyond the data manifold, which may lead to **"manifold intrusion"**. Consider one experiment on MNIST. If we use MixUp to linearly mix two numbers, such as "1" and "5", the generated image will show the characteristics of "8". When the generated "8" **collides with** a real "8" in the data manifold, there will be a problem of manifold intrusion. Since the two samples have similar characteristics, one is the real label and the other is a soft label ("1" and "5"). This will interfere with its ability to understand and classify categories and degrade model performance.

* **Patch-level:** The problem of manifold intrusion also occurs in the patch-level method, termed **"label mismatch."** This occurs when the chosen source patch doesn't accurately represent the source object, leading the interpolated label misleads the model to learn unexpected feature representation. For example, using CutMix to mix images of a cat and a dog. CutMix might select 20% of the background area from the cat image without information about the object (cat). However, their interpolated labels encourage the model to learn both objects’ features (dog and cat) from that training image and degrade model performance.

**The drawbacks of image-level methods:**

Image-level data augmentation increases data diversity by applying label-preserving transformations to the whole image. Notable among these are search-based methods like AutoAugment, RandAugment, and FastAugment. While they improve performance effectively, the **computationally expensive search** for an optimal augmentation policy often exceeds the training process’s complexity. Thus, efforts to minimize the **search space**, **optimize search parameters**, and **uncover potential universal pipelines** are central to the effectiveness of these methods.

In conclusion, we solve these questions by:

* Incorporate **structural complexity** from synthetic data at various levels to produce more diverse images. Our method is **label-preserving**, ensuring it is not affected by manifold intrusion.

* Randomly sample operations from PIL (e.g., brightness, sharpness) and randomly sample strengths to enhance the diversity of training data **without expensive searching**.

* Integrate three levels of data augmentation into a single framework with limited computational overhead, demonstrating that these approaches are complementary and that a unification among them is necessary to achieve robustness.

Due to space constraints, we provide abbreviated references here for clarity and convenience.

[1] Cubuk, Ekin Dogus et al. “AutoAugment: Learning Augmentation Strategies From Data.”

[2] Cubuk, Ekin Dogus et al. “Randaugment: Practical automated data augmentation with a reduced search space.”

[3] Baradad, Manel et al. “Learning to See by Looking at Noise.”

[4] Hendrycks, Dan et al. “PixMix: Dreamlike Pictures Comprehensively Improve Safety Measures.”

[5] Summers, Cecilia and Michael J. Dinneen. “Improved Mixed-Example Data Augmentation.”

[6] Hendrycks, Dan et al. “Unsolved Problems in ML Safety.”

---

### Comment · Area_Chair_TsJd · 2023-08-13
**Please respond to author rebuttals**

Dear Reviewers,

Please respond to authors after carefully reading the author rebuttals and other reviews. If your assessment of the paper changes, please update your score with a short justification for the new rating.

The paper received diverging initial reviews. Please consider discussing with the authors or other reviewers to see whether we can reach a consensus.

Thank you,
AC

---

> ### Author Response · Authors · 2023-08-14
> **Appreciation and understanding during the rebuttal phase**
>
> Dear Area Chair TsJd,
>
> Thank you for your attention and promptness. We understand the workload of reviewers and appreciate their time. We sincerely hope the process continues smoothly.
>
> Best, Authors

---

### Decision · Program_Chairs · 2023-09-21

**Decision:**

Accept (poster)

**Comment:**

The paper proposes IPMix that mixes images at three different levels - image, patch and pixel. This helps the model generalize better of corrupted images. The review ratings are 1 accept, 2 weak accepts, 1 borderline accept and 1 borderline reject. The reviewer who gave borderline reject mentioned that they would like to change it to borderline accept in the reviewer-ac discussion stage. The reviewers agree that the method is easy to follow and the experiment results are strong for both clean and corrupt accuracies. The AC is pleased to recommend acceptance of the paper. The authors should include the comparisons with RandAug and others in the rebuttal to the main paper as one reviewer suggested.